

# Characterization of atmospheric water-soluble brown carbon in the Athabasca Oil Sands Region, Canada.

Dane Blanchard[1], Mark Gordon[2], Duc H. Dang[1,3], Paul A. Makar[4], Julian Aherne[1]

[1]School of the Environment, Trent University, Peterborough, Ontario, Canada K9J 7B8
[2]Earth and Space Science, York University, Toronto, Canada M3J 1P3
[3]Department of Chemistry, Trent University, Peterborough, Ontario, Canada K9J 7B8
[4]Air Quality Research Division, Environment and Climate Change Canada, Toronto and Montreal, Canada

*Correspondence to*: Dane Blanchard (daneblanchard@trentu.ca)

**Abstract**. Extensive industrial operations in the Athabasca Oil Sands Region (AOSR), Alberta, Canada, are a suspected source of
water-soluble brown carbon (WS-BrC), a class of light-absorbing organic aerosols capable of altering atmospheric reaction chemistry and planetary albedo. However, the current understanding of WS-BrC throughout the AOSR is limited, and the primary regional sources of these aerosols are unknown. During the summer of 2021, active filter-pack samplers were deployed at five sites throughout the AOSR to collect total suspended particulate for the purpose of evaluating WS-BrC. Ultraviolet – visible spectroscopy and fluorescence excitation-emission matrix (EEM) spectroscopy, complimented by Parallel Factor Analysis
(PARAFAC) modelling, were employed for sample characterization. Combined EEM-PARAFAC analysis identified three fluorescent components (fluorophores), including one humic-like (C1) and two protein-like (C2, C3) substances. Sites near oil sands facilities and sample exposures receiving atmospheric transport from local industry (indicated via back-trajectory analysis) displayed increased C1 and C3 fluorescence; moreover, both fluorophores were positively correlated with particulate elements (i.e., vanadium, sulphur) and gaseous pollutants (i.e., nitrogen dioxide, total reduced sulphur) indicative of oil sands emissions.
Fluorophore C2 presented high emission intensity at near-field sites and during severe wildfire smoke events, while positive correlations with industry indicator variables suggested C2 was likely reflective of both wildfire-generated and anthropogenic WS-BrC. These results demonstrated that combined EEM-PARAFAC analysis is an accessible and cost-effective tool that can be applied to monitor industrial WS-BrC in the ASOR.

## 1 Introduction

Water-soluble brown carbon (WS-BrC) represents a class of organic aerosols capable of absorbing light within the ultraviolet (UV) and visible (Vis) wavelength ranges (Fuzzi et al., 2002; Lin et al., 2012; Stone et al., 2009). Chemical species within the WS-BrC fraction are classified as chromophoric dissolved organic matter (CDOM), which generally consists of polymerized, aromatic, humic-like substances (HULIS) and lower molecular-weight protein-like substances (PRLIS). Both HULIS and PRLIS represent broad chemical groups named according to their chemical and optical similarity to humic matter and proteins found in terrestrial,
aquatic, and atmospheric environments (Dey and Sarkar, 2024; Duarte et al., 2007; Graber and Rudich, 2006; Hudson et al., 2007; Laskin et al., 2015).

Owing to their light-absorbing properties and capacity as cloud condensation nuclei (CCN), WS-BrC particles can influence planetary albedo and solar radiation budgets (Facchini et al., 1999). Moreover, soluble BrC has been found to play an important role in atmospheric reaction chemistry (Jiang et al., 2012), and can be detrimental to human health (Ma et al., 2019; Maulderly and
Chow, 2008). Biomass and fossil fuel combustion are the dominant sources of atmospheric WS-BrC, including HULIS and PRLIS



(Claeys et al., 2012; Gao et al., 2006; Wu et al, 2019); however, biogenic emissions, soil dust, and industry can all contribute to primary and secondary formation of these organic aerosols (Spranger et al., 2020; Wu et al., 2019).

Industrial operations in the Athabasca Oil Sands Region (AOSR) in north-eastern Alberta, Canada, are a substantial source of atmospheric particulate matter (PM) (Giesy et al., 2010; Landis et al., 2017; Liggio et al., 2016). Open pit mining activities, haul road dust, dry tailings, vehicle exhaust, and upgrading facilities are sources of organic–containing PM (Giesy et al., 2010; Zhang et al., 2016), while volatile precursor emissions from flue gas stacks and exposed bitumen contribute to secondary organic aerosol formation (Liggio et al., 2016). Organic PM generated from oil sands (OS) activities consists of a diverse range of chemical species, including polycyclic aromatic compounds (PAC: Bari & Kindzierski, 2018; Liggio et al., 2016; Liggio et al., 2017), oxygenated polycyclic aromatic hydrocarbons (Jariyasopit et al., 2021), naphthenic acids (Yassine and Dabek-Zlotorzynska, 2017), and organic matter linked to surface-soil and overburden dust (Wang et al., 2015). An unresolved fraction of these emissions contains relatively water-soluble species (Arp et al., 2014; Josefsson et al., 2015; Marentette et al., 2015), which can contribute to the WS-BrC fraction, partially due to their conjugated structures (Stedmon and Nelson, 2015).

Relatively little is known regarding the abundance and composition of WS-BrC in the AOSR airshed, nor is it well understood how local industry contributes to the CDOM fraction. Considering the abundance of sources in the region, OS emissions likely influence WS-BrC variability. The established toxicity and environmental mobility of water-soluble organic pollutants (Lemieux et al., 2008; Lundstedt et al., 2007) provide a clear incentive to monitor these species throughout the AOSR; however, the diversity of chemical formulae presents a challenge to regional monitoring efforts, due to the high operational costs of typical mass spectrometry methods (Frysinger et al., 2003).

Ultraviolet-visible (UV-vis) spectroscopy is commonly employed to characterize the bulk properties of environmental CDOM. Moreover, a sizable fraction of WS-BrC consists of fluorescent dissolved organic matter (Laskin et al., 2015) which can be evaluated through the application of excitation emission matrix (EEM) spectroscopy. Complex EEM datasets can undergo processing via Parallel Factor Analysis (PARAFAC); a multivariable model that can identify fluorescent components (fluorophores) reflective of organic compounds with shared optical properties. Fluorescence EEM spectroscopy and PARAFAC modelling has been successfully used to identify WS-BrC originating from a variety of biogenic and anthropogenic sources (Harsha et al., 2023; Wu et al., 2019; Zito et al., 2019).

In this study we evaluated WS-BrC quality in the AOSR and assessed the influence of OS activities over the organic PM fraction through the application of UV-vis and fluorescence spectroscopy techniques. During the Summer of 2021 (July 19–August 10), active filter-pack samplers were deployed at five sites throughout the study region to collect atmospheric PM for optical and chemical analysis. Fluorescent analysis of local reference materials and back-trajectory modeling were used to assist with source identification, while partial least squares regression (PLS-R) analysis was employed to characterize individual fluorophores generated by the PARAFAC model.

## 2 Methods

### 2.1 Study area and sampling sites

The AOSR is situated within a Boreal Plains ecozone, where peatlands and boreal forests are the dominant landscape features. Largely because of its northern inland geographical location, the region typically experiences warm summers and cold dry winters (ECCC, 2024). Beneath the boreal landscape lies the Athabasca oil sands deposit, the world's largest known bitumen reserve,





which covers a surface area of ~ 140 200 km$^2$. Oil and gas companies such as Syncrude, Suncor, and Imperial Oil have established industrial facilities over these deposits to extract bitumen and upgrade it to refined material such as crude oil. The major OS facilities are located along the east and western banks of the Athabasca River (which flows North through the AOSR), as the

Athabasca bitumen layer is close to the surface in these areas. The largest municipalities in the region are Fort McMurray and Fort McKay; the former is located ~50 km south of major OS operations while Fort McKay sits along the western bank of the Athabasca River, situated between several major industrial facilities (Figure 1).

The Wood Buffalo Environmental Association (WBEA; https://wbea.org) operates an extensive network of continuous air-quality monitoring stations and atmospheric deposition monitoring plots throughout the AOSR. Between July 19$^{th}$ to August 10$^{th}$, 2021,

active filter-pack systems for this study were co-deployed at five WBEA sites located throughout the AOSR to collect back-to-back 48 h exposures (Figure A1). Two of these fence-line sites (Mildred Lake deposition plot [DP2050]; Bertha Ganter station [AMS01]) were located in close proximity (DP2050 < 1 km, AMS01 < 5 km) to OS facilities (consisting of tailings ponds, open pit mines, haul roads, upgrading facilities), a second set of stations (Patricia McInnes [AMS06]; Athabasca Valley [AMS07]) were situated in suburban/community locations > 50 km south of major OS operations, while a final background station (Stoney

Mountain [AMS18]) was located ~ 150 km south of OS operations (Figure 1). Differences in observations between the stations thus allow us to infer sources associated with OS versus other activity, in addition to backtrajectory analysis. The selected WBEA stations continuously monitor a variety of air quality and meteorological data, including sulphur dioxide (SO$_2$), total reduced sulphur (TRS), nitrogen oxide (NO), nitrogen dioxide (NO$_2$), total hydrocarbons (THC), carbon monoxide (CO), fine particulate matter (PM$_{2.5}$), wind speed, wind direction, temperature, and relative humidity.

During the year of 2021, 4.30 x 10$^6$ hectares of land across Canada were burned by wildfire compared to the 2013–2022 ten-year annual average of 2.73 x 10$^6$ hectares (source: Canadian Interagency Forest Fire Center Inc.). As a result of increased fire magnitude, substantial portions of western Canada were periodically blanketed in smoke plumes during the study period. Wildfire emissions are a substantial source of atmospheric PM and BrC (Chen and Bond, 2010; Reid et al., 2005), which can contain aromatic pollutant species with fluorescent properties (Veselovskii et al., 2020). To account for the potential influence of wildfire

emissions over observed fluorescence, the temporal variability of PM$_{2.5}$ was assessed at each WBEA monitoring station to identify sampling periods of high fire smoke transport.



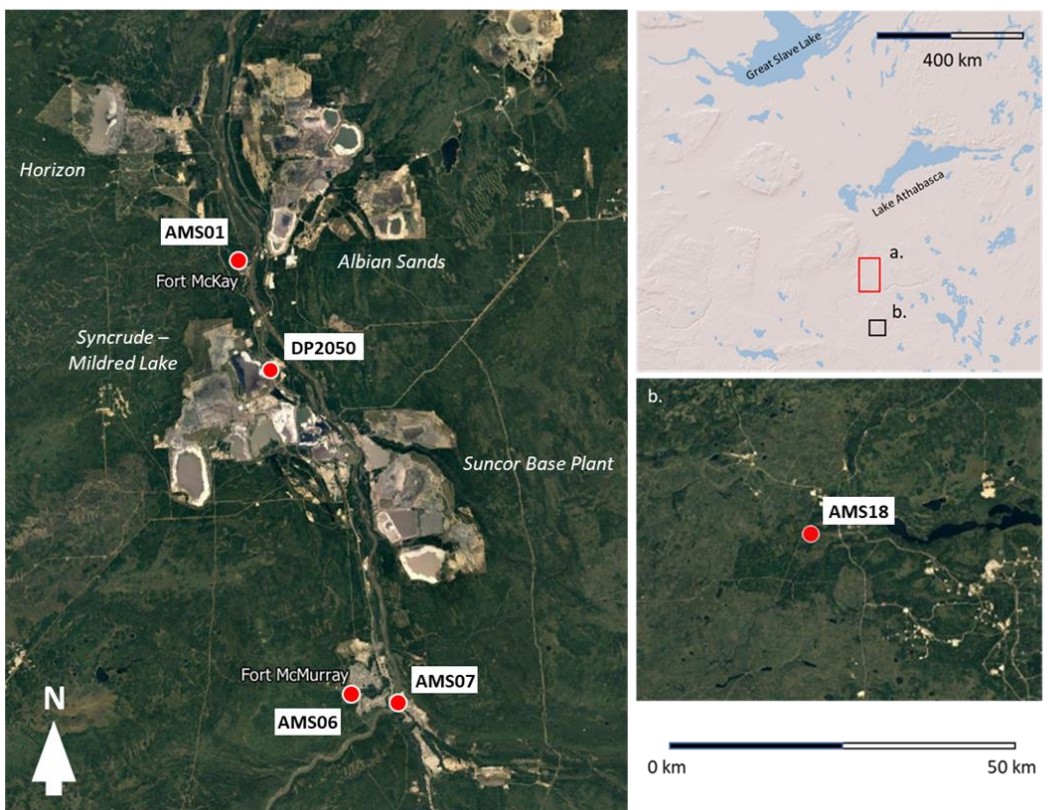

**Figure 1: Distribution of active sampler sites (red circles) throughout near/mid-field (a) and far-field (b) study regions. Nearby communities (Fort McMurray, Fort McKay) and major active oil sands facilities are shown (© Google Maps 2024). The map on the top-right displays the location of the two regions on a broader spatial scale (© ESRI 1995-2024).**

## 2.2 Active sampling

Active samplers consisted of a 15 liter per minute (L min⁻¹) vacuum pump connected to an in-line flowmeter and filter-pack cartridge inlet. These systems were directly powered through an AC outlet available at the WBEA monitoring stations, while a 12 V marine battery served as the power supply for a 7.5 L min⁻¹ vacuum pump at the off-grid DP2050 deposition plot.

The filter-pack cartridges used in this study were obtained through the Norwegian Institute for Air Research (NILU; www.nilu.no) and consisted of two threaded compartments that housed pre-combusted quartz fiber filter (47 mm diameter, 0.3 µm aerosol retention) intended to collect total suspended particulate (TSP). After each exposure, filters were placed in a petri-dish, triple sealed in Ziploc® plastic bags and stored at – 20 °C until analysis. In total, 44 filter-pack samplers were successfully deployed (48 h exposures) during the field campaign, which broken-down by site included: 11 at AMS01 and Mildred Lake, 10 at AMS06, 8 at AMS18 and 4 at AMS07, respectively (Figure A1). During the study period, five of the forty-six samples were collected as duplicates through the deployment of two sampling units. Five additional quartz fiber filters were allocated as field blanks during the study period and were stored like exposed filters.



Prior to the field survey all components of the filter-pack cartridge were individually rinsed in reverse osmosis water and subsequently soaked in a 2 M hydrochloric acid (HCl) bath for 12 hours, after which the sampler parts were rinsed in ultrapure lab water (Milli-Q). In the field, the filter-pack cartridges were soaked in baths of Milli-Q water between exposures. To control for potential filter contamination between exposures, field blanks were loaded into (as if to collect a sample) and subsequently removed from select washed cartridges.

Following exposure, a 20 mm diameter sub-section of the filter was removed via circular punch, and subsequently extracted in conical vials containing 50 ml of Milli-Q water. Sample vials were inverted three times, placed in an ultrasonic bath for 1 hour, and filtered (nylon; 0.45 μm). Filtered samples were stored at ~ 4 °C and underwent optical analysis within 12 hours of initial extraction. A 0.5 ml aliquot was used for determination of anionic species (Cl$^-$, NO$_3^-$, SO$_4^{2-}$) via ion chromatography (Dionex ICS-1100) while a separate 10 ml aliquot was acidified (2 % HNO$_3$) and sent to the Water Quality Center at Trent University (Peterborough, Ontario, Canada) for determination of dissolved elements (Na, Mg, P, S, K, Ca, V, Cr, Mn, Fe, Zn, Sr) using an Agilent 8800 triple quadrupole inductively coupled plasma mass spectrometer. Additional details regarding instrument configuration and measurement parameters are available in the appendix section A1. A filtered 20 ml aliquot was used for determination of water-soluble organic carbon (WSOC), dissolved inorganic carbon (DIC), and total dissolved nitrogen (TDN) via Shimadzu TOC-V analyzer. The remaining filtered solution was adjusted to pH ~ 6.5 using 0.1 M sodium hydroxide (NaOH) or HCl in preparation for optical analysis. Blank samples were used to determine method detection limits (MDL), which were calculated as: standard deviation (blank samples) × t-value 99.0 % confidence critical value (3.747).

Measured analytes from the filter extracts were converted to time-integrated atmospheric concentrations using the following equation:

$$C = \frac{(C_e - C_b) * V_e}{V_a} * A_f * 1000 \tag{1}$$

Where $C$ is the atmospheric concentration (ng m$^{-3}$), $C_e$ is the analyte concentration (μg L$^{-1}$) within the filter extraction solution, $C_b$ is the analyte concentration within the field blank extraction solution, $V_e$ is the extraction solution volume (L), $V_a$ is the volume of air sampled (m$^{-3}$), and $A_f$ is the surface-area ratio (5.52) to account for the 20 mm filter cut-out.

### 2.3 UV-Vis spectroscopy

Absorbance was measured along a range of wavelengths (λ) between 200 to 800 nm (1 nm intervals) using a Cary 100 UV-vis spectrophotometer from Agilent Technologies. Filter extracts were allowed to warm to room temperature (~ 21 °C), and subsequently placed in a 1 cm (3.5 ml) Hellma® Suprasil® quartz cuvette (Hellma 101-QS) for analysis. A blank sample of Milli-Q water was first used to establish an instrument baseline. To clean, cuvettes were soaked in a 2 M HCl bath for ten minutes followed by subsequent rinses in Milli-Q water, 95% ethyl alcohol, and a final rinse in Milli-Q. The aerosol light absorption coefficient (Abs$_\lambda$, m$^{-1}$) at wavelength λ was calculated by the equation (Wu et al., 2019):

$$Abs_\lambda = (a_\lambda - a_{700}) \times \frac{V_w}{V \times l} \times \ln(10) \tag{2}$$

Where $V_w$ refers to the volume (ml) of the filter extraction, $V$ is the volume of air sampled (m$^3$) during the field exposure, $l$ is the optical path length (cm), while $a_\lambda$ and $a_{700}$ are the absorbance at wavelengths $\lambda$ and 700 nm, respectively. Absorbance at 700 nm is used as a reference to account for baseline drift. The light absorption coefficient at 365 nm (Abs$_{365}$) is frequently referenced as a proxy for BrC in the literature (Laskin et al., 2015; Wu et al., 2019), and is subsequently used to calculate the mass absorption efficiency (MAE$_{365}$, m$^2$ g C$^{-1}$) via the following equation (Wu et al., 2019):



$$MAE_{365} = \frac{Abs_{365}}{C_a} \tag{3}$$

Where $C_a$ is the atmospheric concentration of WSOC (µg C m⁻³), obtained using Eq. 1.

**2.4 Fluorescence spectroscopy and PARAFAC**

Fluorescence analyses was conducted using a Cary Eclipse spectrophotometer set to the three-dimensional EEM scan and signal to reference (S/R) acquisition modes. Sample aliquots were placed in a 1 cm (3.5 ml) quartz cuvette (Hellma 101-QS) for analysis. Excitation ($\lambda_{Ex}$) and emission ($\lambda_{Em}$) scans (both measured at 5 nm intervals) ranged from 200–450 nm and 250–600 nm,

respectively. A Milli-Q water blank was first measured to zero the fluorometer, then subsequent blanks were scanned every five samples. During each day of analysis, a set of quinine sulphate standards were prepared to create a calibration curve (1–100 ppb quinine sulphate in 0.5 M sulphuric acid) to convert fluorescence intensity measurements into quinine sulphate units (QSU) (Sui et al., 2017). To minimize the potential influence of the inner filter effect (IFE) over the fluorescence spectrum, all filter extracts were confirmed to have an absorbance < 0.05 at 270 nm (Trubetskoj et al., 2018).

Fluorescence data was processed via the R package "staRdom", which was used for blank and spectrum correction (Pucher et al., 2019). Inner filter effect corrections were applied to individual EEMs using the built-in IFE function and corresponding sample absorbance values (Shimadzu spectrophotometer). Rayleigh and Raman light scattering bands were removed from each sample EEM, and the empty cells infilled using the "eem_interp" function. The fluorescence intensity data was then divided by the corresponding extraction solution and air volume to convert QSU to QSU per cubic meter (QSU m⁻³) (Deng et al., 2022). The EEM

dataset displayed limited fluorescence within the 540–600 emission and 400–450 excitation wavelength ranges, thus these regions were removed prior to PARAFAC modeling. Using the staRdom package, a PARAFAC model was employed to decompose the three-dimensional EEM data array ($X_{ijk}$) into a score matrix (A) and two loadings' matrices (B and C) with corresponding tri-linear elements $a_{if}$, $b_{jf}$, and $c_{kf}$ (Bro 1997):

$$x_{ijk} = \sum_{f=1}^{F} a_{if} b_{jf} c_{kf} + e_{ijk} \tag{4}$$

Score values in $a_{if}$ represent the relative fluorescence intensity of fluorophore $f$ in sample $i$, the elements $b_{jf}$ and $c_{kf}$ represent the modelled emission ($j$) and excitation ($k$) loadings (i.e., wavelength coordinates) of $f$, while $F$ is the total number of modelled fluorophores. Utilizing the residual element $e_{ijk}$, the PARAFAC model applies an alternating least-squares optimization technique to find the function that best approximates the dataset (Bro, 1997).

The appropriate number of modelled components (model rank) was selected based on four validation tools, including: 1) visual

peak inspection, 2) EEM residual plots, 3) split-half analysis, and 4) core consistency (CC). Visual inspection provides an initial check to confirm the modelled components present well-defined emission peaks, which suggests that the spectra represent a distinct fluorophore or a group of similarly fluorescing species. The model residuals corresponding to each sample ($e_{ijk}$) can be plotted as a function of $\lambda_{Ex}$ and $\lambda_{Em}$. These plots can be visually inspected to identify potential peaks, where a high abundance of such features across the dataset would indicate that the model rank is not representative of the array (Driskill et al., 2018; Murphy et al., 2014).

Split-half analysis involved division of the EEM dataset into a set of four equal-sized groups, where independent PARAFAC models were generated for each subgroup and subsequently compared to measure similarity via Tucker Congruence Coefficient (TCC; Pucher et al., 2019). A high TCC similarity score (~1) signals that the model is representative of the true variation within the fluorescence dataset. The model rank with the highest TCC score is most likely the appropriate selection (Bro, 1997; Stedmon et al., 2003). Core consistency is a measure of the regression between the three-way array of the PARAFAC model and the core

array of a Tucker3 model generated from the corresponding EEM data. A CC score between 90% to 100% indicates a strong



model, while values ≤ 50% suggest a problematic output. Generally, the largest model rank prior to a dramatic decline in CC should be selected (Bro and Kiers, 2003).

The fluorescent components generated from the PM samples were compared against similar PARAFAC models published in the online database OpenFluor to assist with fluorophore characterization and source identification (Murphy et al., 2014). Model

components with high similarity scores to these database models (> 0.95) along both excitation and emission spectrum were selected for comparative analysis. The sample score values assigned to individual fluorescent components ($a_{if}$) were reported according to maximum fluorescence ($F_{MAX}$, QSU m$^{-3}$). To account for the potential influence of bulk DOM fluorescence over individual fluorophore emission intensity (due to variations in DOM concentration between samples), modelled component $F_{MAX}$ values were divided by sample WSOC concentration (µg C m$^{-3}$) to normalize component fluorescence.

**2.5 Environmental refer1ence materials**

A variety of environmental reference materials representative of dominant TSP emission sources from the OS were collected for EEM analysis, including a) dust from an unpaved road near observation site AMS01 (composite grab-sample), b) dust from a heavily trafficked unpaved road near observation site DP2050 (composite grab-sample), c) raw bitumen ore from the Athabasca Deposit and, d) dry mature fine tailings from the AOSR (MFT; composite industrial sample; Innotech Alberta). Dry road dust and

MFT were sieved to < 2 mm to remove coarse material (Lanzerstorfer and Logiewa, 2019), while raw bitumen was too viscous to sieve. Each reference material was extracted in Milli-Q water and diluted to achieve a WSOC concentration of ~ 1.0 mg C L$^{-1}$ and an absorbance < 0.05 at 270 nm (to match the filter extracts). Sample extracts were adjusted to a pH ~ 6.5 using 0.1 M NaOH or HCl and then stored at ~ 4 °C prior to optical analysis.

**2.6 Data analysis**

Variability in sampler observations was assessed by calculating the relative percent difference (RPD, %) from duplicate filter-pack exposures. Data displayed a non-normal distribution as determined via Shapiro-Wilk test (α < 0.05); thus, non-parametric statistical tests were employed for subsequent analysis. Median values were used as a measure of central tendency, while data variability was evaluated using normalized median absolute deviation. Bivariate associations between measured TSP analytes and continuous pollutants were evaluated via Spearman's Rank Order correlation test, and the Kruskal-Wallis test was used to determine significant

differences between sampling locations.

Partial least squares regression (PLS-R) analysis was applied to assist with fluorescent component characterization (Kothawala et al. 2014). Partial least squares analysis is a multivariable regression technique that is effective at decomposing large numbers of colinear predictor variables (X) into a smaller set of uncorrelated orthogonal components (Eriksson et al., 2013). The PLS-R model attempts to explain the maximum variability between X and response variables (Y) through a least-squares regression method.

Modelled fluorescent component values (QSU m$^{-3}$) were assigned as Y variables while additional elements, WSOC, DIC, and TDN were designated as X. The appropriate number of components was determined via internal cross-validation, where a random subset of samples (~ 10 % of dataset) was removed from the training dataset to evaluate the model. The resulting comparisons were evaluated based on model rank cumulative goodness of fit (explained variation; $R^2Y$) and cumulative goodness of prediction ($Q^2Y$). The relative influence of X variables over each component was interpreted via variable influence on projection scores

(VIP), where values ≥ 1 signified a highly influential predictor variable (Eriksson et al., 2013). The spatial proximity of variable loadings served as a visual indicator of X and Y variable associations, while variable distance from the origin indicated the level of correlation with each component.



**2.7 Trajectory analysis**

The atmospheric transport model, HYSPLIT (Stein et al., 2015), was applied to evaluate source-receptor relationships within the study region. Meteorological data available through the Global Data Assimilation System was employed to generate bihourly backward wind trajectories (arrival height 10 m above ground level) spanning the duration (~48 h) of active sampler exposures at individual sites (i.e., 24 back-trajectories per exposure). The resulting trajectory data were overlain onto OS facilities (polygons representing tailings, open mine pit, plant site boundaries) to explore the influence of industrial sources over observed TSP fluorescence. Using these spatial data, the relative frequency (%) of back-trajectories intersecting OS-facility boundaries ($T_{OS}$)

during each exposure was calculated to further explore source-receptor relationships. The frequency method counts the number of times air trajectories intersect individual grid points ($0.5 \times 0.5$ degrees) in the model domain, and then normalizes that sum value by the total number of modelled trajectories (%). It should be noted that the trajectory estimates do not account for the influence of vertical mixing and lateral diffusion over pollutant transport.

Through the Canadian National Fire Database, spatial data representing the boundaries of Canadian wildfires active during the

study period (July 19–August 10) were used to evaluate the potential influence of pyrogenic sources over observed fluorescence. Using the same GDAS meteorological datasets, additional HYSPLIT backward trajectory frequency plots were modelled to evaluate atmospheric transport patterns during multi-day wildfire smoke episodes.

**3 Results**

**3.1 Meteorology and continuous pollutant summary**

Limited variation in relative humidity occurred between monitoring stations during the study period, where median values ranged from 69.8 % to 71.8 % (Table 1). The AMS01, AMS06 and AMS07 stations experienced prominent winds blowing from a range of directions (excluding NE – E) during the study period, while S – W winds were dominant in AMS18. Wind speeds were higher at AMS06, while colder temperatures were recorded at AMS18. Moreover, the background station received a larger volume of precipitation compared with AMS01. Further, AMS01 experienced significantly higher CO, $NO_2$, $SO_2$, and TRS (Kruskal-Wallis;

$\alpha < 0.05$) owing to its proximity to OS operations, while the lowest concentrations (with exception of CO) were recorded at the more distant downwind site AMS18 (Table 1). Ambient $PM_{2.5}$ was notably elevated at AMS18 with concentrations similar to AMS06 and AMS07.

Evaluation of WBEA station $PM_{2.5}$ timeseries data identified two periods of elevated concentrations. The first and most intense (hourly maximum $PM_{2.5} > 140$ µg m$^{-3}$) event occurred between July 19–22 while the second event ranged from July 31–August 6

(hourly maximum $PM_{2.5} \sim 50$ µg m$^{-3}$) (Figure A2). These high $PM_{2.5}$ events occurred during relatively similar periods between monitoring stations (including AMS18), suggesting a shared regional source. Trajectory frequency analysis indicated that extensive wildfires in northern Saskatchewan were the primary source of smoke ($PM_{2.5}$) in the study region (Figure A3).

The $T_{OS}$ analysis demonstrated that the relative influence of OS emissions was expected to be strongest at the DP2050 (79.2 %) and AMS01 (43.8 %) sites, followed in descending order by AMS06, AMS07, and AMS18. Differences in our analysis at each of

these sites thus can be used to infer source-related differences in composition (Table 1). Comparison of HYSPLIT trajectories against corresponding continuous station data displayed a general agreement between modelled and observed wind direction between exposure periods (Figure A4). Bivariate analysis demonstrated strong positive correlations ($R_S = 0.83 - 0.92$) between $T_{OS}$ and continuous pollutant species reported through WBEA (exposure average NO, $NO_2$, $SO_2$, TRS, THC; Table A1),





emblematic of emission transport from OS facilities. Frequency estimates displayed weaker agreement with average $PM_{2.5}$ and CO

($R_S < 0.50$), likely because of the influence of non-OS sources, namely wildfire smoke.

**Table 1: Summary of geographical (elevation), meteorological (wind speed, temperature, relative humidity, precipitation), air quality ($PM_{2.5}$, CO, NO, $NO_2$, $SO_2$, TRS, THC), and transport frequency ($T_{OS}$) variables at the co-located WBEA monitoring stations (ordered according to latitude [N to S]) during the study period (July 19–August 10, 2021). Both median and normalized median absolute deviation (%; in-parenthesis) values for each variable and station are displayed.**

|  | AMS01 | AMS06 | AMS07 | AMS18 |
|---|---|---|---|---|
| Latitude | 57.189428 | 56.751378 | 56.73344 | 55.621408 |
| Longitude | -111.640583 | -111.476694 | -111.39047 | -111.172686 |
| Elevation (m) | 270 | 362 | 497 | 673 |
| Relative humidity (%) | 70.7 (21.7) | 69.8 (20.3) | 71.8 (17.1) | 71.3 (24.6) |
| Wind speed (km hr$^{-1}$) | 6.2 (46.1) | 8.4 (43.1) | 6.7 (44.0) | N/A |
| Temperature (°C) | 17.5 (18.5) | 17.5 (17.9) | 18.3 (16.1) | 16.6 (21.9) |
| Precipitation (mm) | 45.3 | N/A | N/A | 48.0 |
| $PM_{2.5}$ (µg m$^{-3}$) | 10.27 (63.8) | 8.85 (61.2) | 9.08 (58.6) | 9.08 (55.3) |
| CO (ppm) | 0.40 (20.0) | N/A | 0.22 (22.7) | 0.28 (42.9) |
| NO (ppb) | 0.67 (25.4) | 0.42 (50.0) | 0.63 (68.0) | 0.34 (14.7) |
| $NO_2$ (ppb) | 3.28 (75.1) | 1.48 (77.7) | 2.23 (55.3) | 0.57 (38.6) |
| $SO_2$ (ppb) | 0.67 (11.9) | 0.27 (37.0) | 0.17 (29.4) | 0.09 (77.8) |
| TRS (ppb) | 0.54 (22.2) | 0.31 (19.4) | 0.13 (61.5) | 0.09 (44.4) |
| THC (ppm) | 2.11 (6.64) | 1.99 (2.5) | 2.09 (4.3) | 1.91 (1.5) |
| $T_{OS}$ (%) | 43.8 | 22.9 | 12.5 | 0.0 |


## 3.2 Active sampler TSP summary

Throughout the study, the DP2050 sampler measured elevated concentrations of DIC, $NO_3^-$, $SO_4^{2-}$, Na, P, S, V, Fe, and Sr compared with the other study sites, while most variables (with exception of WSOC and TDN) were depressed at AMS18 (Table 2). Measured WSOC and $Abs_{365}$ were significantly higher at the nearfield site DP2050 compared to the remaining stations (Kruskal-Wallis; α <

0.05), while $MAE_{365}$ values were similar across all sampling locations (Kruskal-Wallis; α > 0.05). Dissolved V, Sr, S, and Na were strongly (positively) correlated ($R_S > 0.77$) with $T_{OS}$ estimates, while WSOC and $Abs_{365}$ displayed weak positive agreement ($R_S \geq$ 0.45) (Table A2) with $T_{OS}$. Both WSOC and absorbance values were occasionally elevated during individual exposures (i.e., AMS18, August 5–7, WSOC = 7.2 µg C m$^{-3}$, $Abs_{365}$ = 7.7 m$^{-1}$) receiving limited industrial but high wildfire atmospheric transport (Figure A5), while $MAE_{365}$ displayed limited patterns in relation to wildfire smoke events. Moreover, WSOC and $Abs_{365}$ were

moderately correlated (Spearman α < 0.05, $R_S$ = 0.30).






**Table 2: Summary of active sample variables (with corresponding MDL; ng m⁻³) measured during the study period (July 19–Aug 10) at each monitoring location. Median (and normalized median absolute deviation in-parenthesis) for each variable and site are shown. Sampling locations are ordered according to latitude (N to S).**

| Analyte (ng m⁻³) | AMS01 (n = 9) | DP2050 (n = 11) | AMS06 (n = 8) | AMS07 (n = 3) | AMS18 (n = 7) | MDL (ng m⁻³) |
|---|---|---|---|---|---|---|
| WSOC | 3907.4 (33.0) | 6156.1 (21.6) | 3329.2 (14.5) | 7667.0 (63.7) | 4091.6 (28.8) | 6.3 |
| DIC | 469.7 (15.8) | 1047.4 (17.3) | 476.8 (25.7) | 979.8 (58.7) | 461.8 (41.5) | 20.6 |
| TDN | 475.9 (75.8) | 883.4 (42.7) | 372.1 (50.2) | 1569.9 (63.3) | 732.9 (17.8) | 1.9 |
| $NO_3^-$ | 325.7 (55.1) | 845.6 (67.3) | 530.3 (67.0) | 734.9 (94.5) | 109.0 (44.6) | 18.8 |
| $SO_4^{2-}$ | 900.7 (58.0) | 2931.8 (54.6) | 1113.5 (73.5) | 1275.2 (87.8) | 301.2 (12.0) | 3.3 |
| $Cl^-$ | 9.8 (78.7) | 74.9 (47.7) | 81.7 (67.5) | 79.3 (58.2) | 20.2 (42.4) | 2.4 |
| Na | 64.3 (46.1) | 194.4 (44.5) | 58.7 (85.4) | 55.9 (21.4) | 15.7 (44.6) | 1.9 |
| Mg | 52.2 (53.5) | 119.7 (25.7) | 34.1 (22.3) | 122.9 (77.3) | 19.7 (48.0) | 1.6 |
| P | 7.8 (41.7) | 15.1 (12.6) | 9.6 (14.5) | 10.7 (79.4) | 5.0 (30.1) | 0.4 |
| S | 303.6 (32.1) | 710.3 (51.2) | 203.9 (31.0) | 244.8 (30.9) | 200.8 (12.8) | 0.8 |
| K | 89.7 (19.8) | 91.4 (42.4) | 121.5 (39.6) | 100.8 (51.2) | 44.9 (25.7) | 1.1 |
| Ca | 772.2 (76.8) | 1938.4 (22.1) | 592.9 (5.9) | 1558.3 (87.9) | 315.9 (45.1) | 3.3 |
| V | 0.11 (57.9) | 0.20 (30.6) | < 0.1 (78.7) | < 0.1 (78.7) | < 0.1 (69.1) | < 0.1 |
| Cr | 0.8 (58.8) | 1.9 (9.2) | 0.5 (62.8) | 1.8 (93.5) | 0.4 (9.4) | 0.1 |
| Mn | 3.3 (48.9) | 8.5 (30.9) | 4.5 (42.5) | 8.4 (70.8) | 1.1 (34.8) | 0.1 |
| Fe | 3.7 (44.6) | 8.8 (28.6) | 1.9 (50.9) | 2.5 (36.2) | 1.8 (34.9) | 0.2 |
| Zn | 3.4 (30.8) | 5.0 (80.4) | 8.9 (27.8) | 3.8 (34.3) | 2.8 (14.5) | 0.4 |
| Sr | 1.7 (60.4) | 4.8 (22.2) | 1.6 (77.5) | 0.8 (32.1) | 0.2 (191.0) | 0.2 |
| Abs₃₆₅[a] | 4.1 (44.5) | 7.5 (29.8) | 3.1 (28.0) | 4.5 (47.0) | 3.2 (45.1) | 0.2 |
| MAE₃₆₅[b] | 1.1 (40.0) | 1.0 (45.6) | 0.9 (25.7) | 0.9 (23.8) | 1.0 (29.6) | - |

[a] Reported as m⁻¹
[b] Reported as m⁻² g C⁻¹

### 3.3 Fluorescence

The EEMs produced from the TSP samples frequently displayed peak fluorescence (> 30 QSU) within the $\lambda_{Ex}$ and $\lambda_{Em}$ wavelength ranges of 210–350 nm and 390–430 nm, respectively. Moreover, secondary peaks (> 20 QSU) were observed within the excitation range of 200–250 nm, and emission range of 250–400 nm. Fluorescent scans of raw bitumen and dry MFT materials produced strong peaks (> 30 QSU) in the similar region of $\lambda_{Ex}$ of 200-250 nm and $\lambda_{Em}$ of 250-400 nm, and limited fluorescence within the higher range ($\lambda_{Ex}$ of 210-350 nm and $\lambda_{Em}$ of 390-430 nm) (Figure A6). Similar scans of unpaved road dust and sand pit materials displayed comparatively weak fluorescence (≤ 8 QSU) within these EEM regions.

Comparative analysis of multiple PARAFAC models indicated that a model rank of three was most appropriate, that is, the data suggest three distinct classes of organic compounds were present. The three-component model displayed a high degree of fit (0.95), CC (96.2) and split-half validation (0.87) scores. The modelled components displayed clearly defined fluorescent peaks / regions, while no distinct and/or recuring residual peaks were observed within the model output (Figure 2).

The first fluorescent component (C1) presented two emission peaks generally classified as humic acids in standard databases (Table 3) (Graeber et al., 2012; Williams et al., 2013). Several water-quality studies have associated similar peaks with fluorophores of petrogenic and anthropogenic origin (Brünjes et al., 2022; Zito et al, 2019), while previous BrC studies have characterized similar fluorophores as atmospheric HULIS (Dey and Sarkar, 2024; Laskin et al., 2015). Although there were fewer database matches, component two (C2) was similar to amino acid and tyrosine-like fluorescence (Murphy et al., 2011; Pitta and Zeri, 2021), reflective of PRLIS aerosols identified in the literature (Dey and Sarkar, 2024; Laskin et al., 2015). Component three (C3) was identified as PRLIS BrC with spectra similar to amino acid-like (Cawley et al., 2012; Murphy et al., 2008), anthropogenic (Yan et al., 2020), and petrogenic (Whisenhant et al., 2022) fluorescent DOM. It is important to note that none of the models matched via OpenFluor





were based on atmospheric samples, as the database predominantly contains models derived from marine, freshwater, and
terrestrial environments.

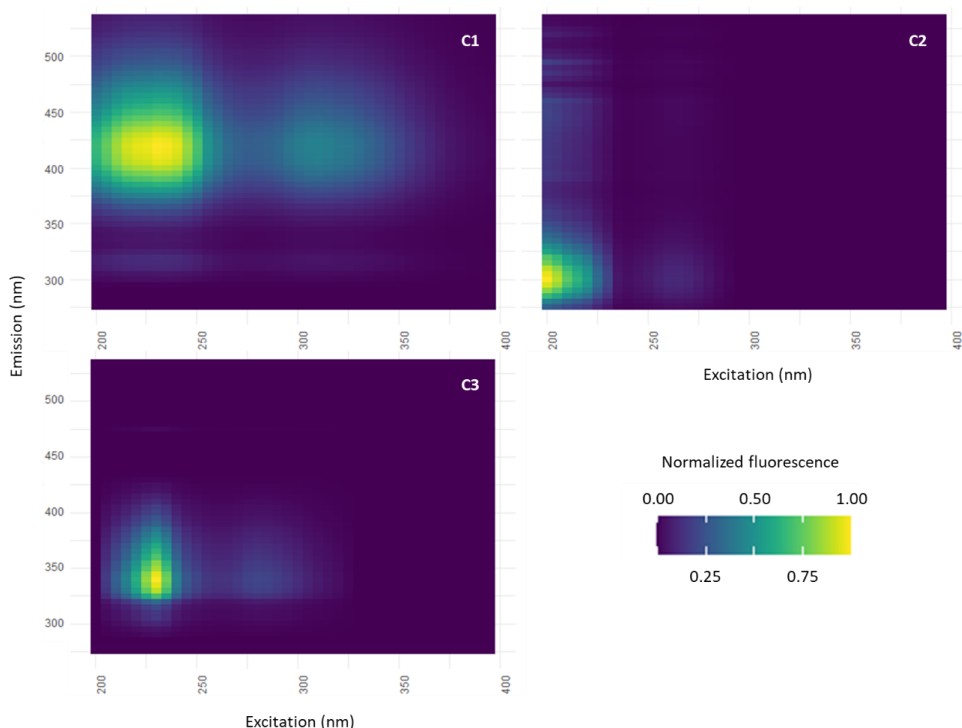

**Figure 2: Visualization of the three PARAFAC components (C1–C3) generated from the combined TSP EEM sample set. Emission intensity values are normalized according to maximum component fluorescence.**







**Table 3: Comparison of PARAFAC model components (C1–C3) against the OpenFluor[a] online database. The excitation ($\lambda_{Ex}$) and emission ($\lambda_{Em}$) wavelength coordinates of each fluorophore, the number of highly similar components (r > 0.98, r > 0.95), general fluorophore classifications based on the cited studies, and corresponding references (with modelled component number) are shown.**

| Component ($\lambda_{Ex}$, $\lambda_{Em}$) | Openfluor matches (r>0.98, r>0.95) | Fluorophore Characterization | Reference and PARAFAC Component |
|---|---|---|---|
| C1 $\lambda_{Ex}$: 230 & 315 nm $\lambda_{Em}$: 420 nm | 83, 117 | Humic acid | Graeber et al., 2012 (C1); Williams et al., 2013 (C2); Yamashita et al., 2011 (C1) |
| | | Petrogenic | Zito et al., 2019 (C1), Brünjes et al., 2022 (C4) |
| | | Anthropogenic | Vines and Terry, 2020 (C1) |
| C2 $\lambda_{Ex}$: 200 nm $\lambda_{Em}$: 300 nm | 2, 11 | Amino acid / tyrosine | Murphy et al., 2011 (C5); Pitta and Zeri, 2021 (C4); Weigelhofer et al., 2020 (C4) |
| C3 $\lambda_{Ex}$: 235 nm $\lambda_{Em}$: 335 nm | 19, 37 | Amino acid | Cawley et al., 2012 (C5); Murphy et al., 2008 (C7) |
| | | Anthropogenic Petrogenic | Yan et al., 2020 (C3); Whisenhant et al., 2022 (C2) |

[a] https://openfluor.lablicate.com

Median normalized $F_{MAX}$ (and normalized median absolute deviation, %) values for C1–C3 were 6.0 (100.0 %), 34.6 (39.5 %), and 10.4 (66.9 %) QSU m$^{-3}$, respectively, demonstrating that the C2 amino acid-like component was the dominant fluorophore. The study average RPD calculated from duplicate exposures was 10.7, 8.5, and 9.6 % for C1–C3, respectively. Both $F_{MAX}$ and WSOC-normalized fluorescence ($C_{WSOC}$) values of C1 and C3 were highest at the near-field industrial sampling locations (DP2050, AMS01), followed by the mid-field urban sites (AMS06, AMS07), while the lowest values were reported at the background

location (AMS18) (Figure 3), indicating a greater likelihood of C1, C3 having OS-related sources. Moreover, measured fluorescence at DP2050 and AMS01 was significantly higher (Kruskal-Wallis; $\alpha < 0.05$) than AMS18, further indicating that OS operations were a dominant source of C1 and C3. The second fluorescent component (C2) displayed a distinct decay in fluorescence intensity as a function of distance from OS operations. Although C2 PRLIS fluorescence was significantly elevated in DP2050 compared to all down-field sites (excluding AMS07 due to small sample size), comparison of the remaining sampling locations

(including AMS18) revealed insignificant differences in measured fluorophore intensity. High relative fluorescence at the background station suggested that a broader, regional (non-OS) source contributed to C2 in the study region.



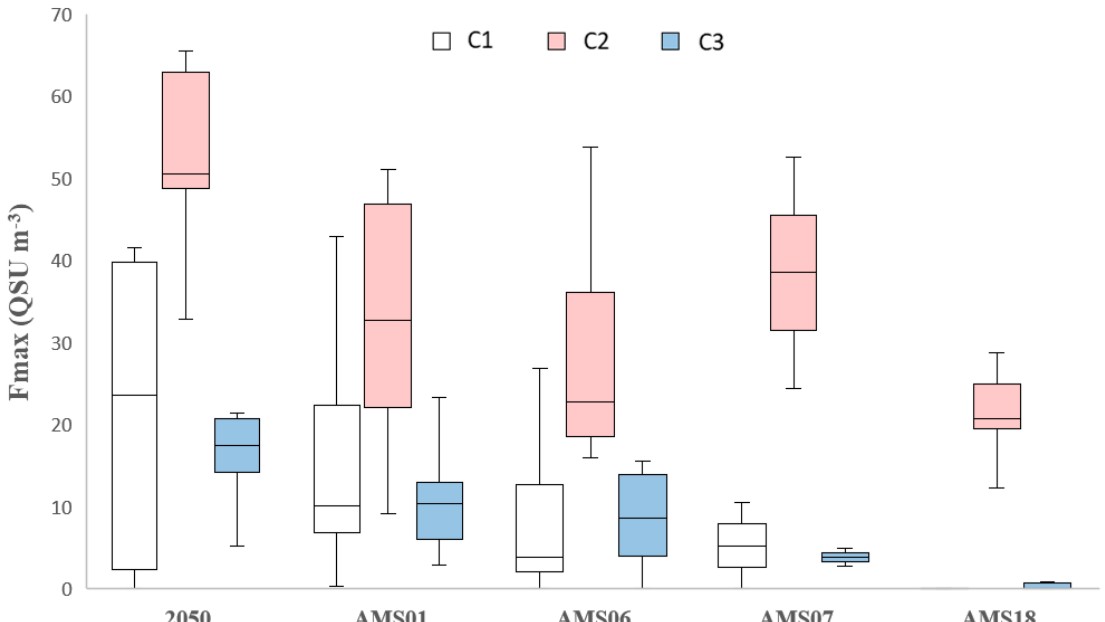

**Figure 3: Distributions of component (C1–C3) fluorescence (QSU m$^{-3}$) measured at each sampling location during the study period (July 19–August 10).**

Both C1 and C3 displayed strong positive correlations ($R_S = 0.77$ and $0.88$, respectively) with $T_{OS}$ (Figure 4; Table A1), suggesting fluorescence was elevated when samplers received atmospheric transport from OS operations. As specific examples, two Mildred Lake exposures (August 6–8, 8–10) with distinctly high C1 and C3 (> 100 and > 38 QSU m$^{-3}$, respectively) received strong SW winds ($\geq$ 10 km h) that passed over adjacent (~1 km from site) tailings ponds, open pit mines, and active upgrading facilities (Figure 5). High $F_{MAX}$ exposures at AMS06 between August 4–6 coincided with N-E winds passing over upwind facilities (Figure

A7). Conversely, when samplers received atmospheric transport from undeveloped / forested regions, C1 and C3 were markedly lower (Figure 5). Although generally elevated when downwind of OS facilities, C2 was occasionally heightened when sites received non-industry trajectories (Figure A5). This inconsistency is reflected by the comparatively weaker correlation between $T_{OS}$ and C2 fluorescence ($R_S < 0.62$). The WSOC-normalized fluorescence (C1$_{WSOC}$–C3$_{WSOC}$) displayed spatial trends relatively similar to C1–C3 (Figure A8).

Fluorescent components C1 and C3 varied independently of wildfire transport; for example, low C1 and C3 (< 0.5 and 5.2 QSU m$^{-3}$, respectively) were recorded during the high-smoke event at DP2050 (July19–20; Figure A5), while some of the highest $F_{MAX}$ values were recorded during relatively smoke-free exposures at the same site (August 8–10; C1 = 137.6 QSU m$^{-3}$, C3 = 20.1 QSU m$^{-3}$). Alternatively, high C2 concentrations frequently occurred during wildfire events, where back-trajectories displayed winds originating from fire-affected regions in BC or Saskatchewan (Figure A4 and A5).





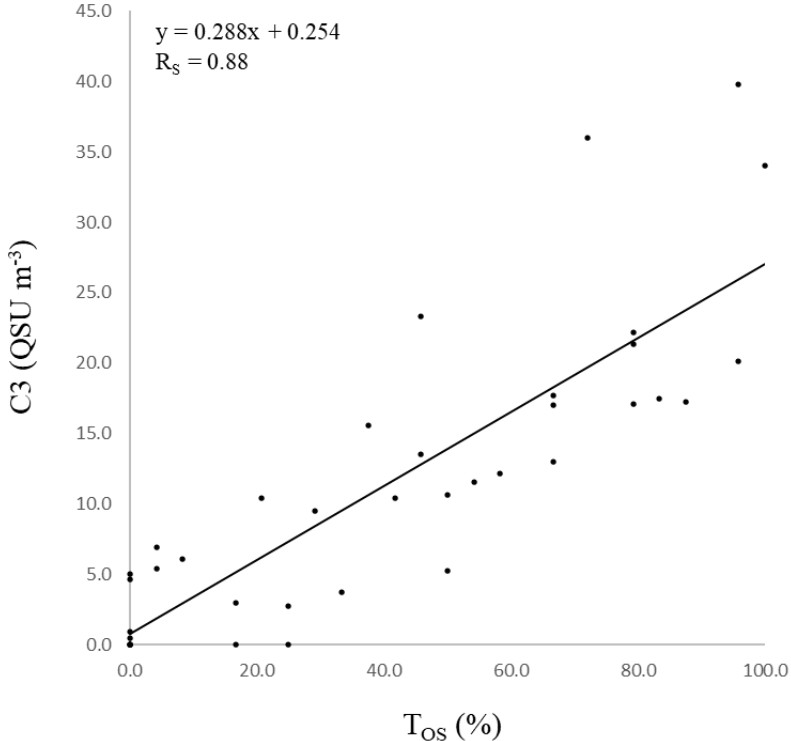


**Figure 4: Bivariate comparison of active sampler measured C3 fluorescence (QSU m$^{-3}$) and corresponding OS trajectory frequency (T$_{OS}$; %) estimates throughout the study period. The line of best fit and corresponding regression equations with Pearson correlation coefficient (R$_S$) is displayed.**





**Figure 5: Spatial distributions of 48 h atmospheric back-trajectories (black lines) converging at sampling locations during selected exposure periods, including: a) DP2050; August 6–8, b) DP2050; July 29–31, c) AMS01; August 4–6 and, d) AMS01; July 21–23. Normalized $F_{MAX}$ values of C1–C3 (QSU m$^{-3}$) measured during each exposure are shown beneath the corresponding trajectory plots. The spatial boundaries of various OS facilities, including wet tailings (yellow), dry tailings (orange), open mine pits (red), and plant sites (purple) are shown (© ESRI 1995-2024).**



### 3.4 Comparison of fluorophores against supplementary TSP variables

Comparison against continuous station data found that C1 and C3 were positively correlated (Spearman; $\alpha < 0.05$) with most pollutant species (Table A1) while C2 displayed comparatively weaker correlations with continuous data. Evaluation of the complete TSP dataset found that C1 was strongly (positively) correlated ($R_s > 0.70$) with $NO_3^-$, $SO_4^{2-}$, Na, Mg, S, V, and Sr; C3 displayed similar trends, with additional correlations with Ca, P, and Mn. Moreover, C2 was moderately correlated with WSOC, DIC, Na, Mg, P, S, Ca, V, Cr, Mn, and Sr (Table A2).

Cross-validation confirmed a two-component PLS-R model that produced a relatively strong $R^2Y$ (cumulative $R^2Y = 0.73$) and the highest cumulative $Q^2Y$ value (0.57). The first modelled component (PLS1, $R^2Y = 0.49$) was primarily defined (VIP > 1) by $NO_3^-$, $SO_4^{2-}$, Na, Mg, P, S, Ca, V, Mn, Sr, and $T_{OS}$, while the second component (PLS2, $R^2Y = 0.24$) was characterized by WSOC, DIC, $SO_4^{2-}$, Na, P, V, and $T_{OS}$. Of these variables, Na and $T_{OS}$ were the most important (VIP > 1.4) to PLS1 and PLS2. Considering certain OS fugitive emissions (e.g., tailings sand and haul road dust) are often Na enriched (Wang et al., 2015), these high VIP scores suggested both dimensions represented industrial sources. Moreover, PLS1 was positively correlated with variables previous linked to crustal (Mg, Ca, Mn) and bituminous (S, Sr, V) sources, which in the case of the AOSR could be reflective of bitumen and overburden excavation (Landis et al., 2012). Concerning PLS2, the positive correlation with Na, $SO_4^{2-}$, and $T_{OS}$, could represent non-crustal industrial sources (Wang et al., 2015), while the negative correlation with WSOC and DIC could reflect non-OS carbonaceous PM emissions. Biplot visualization revealed that C1 and C3 fluorescence were positively correlated with both PLS-R components and plotted near Na (Figure 6); both fluorophores were notably distant from WSOC and $Abs_{365}$. Fluorescent component 2 displayed a strong positive correlation with the first PLS-R component, and was plotted in proximity to WSOC, DIC, Ca, Mn, and $Abs_{365}$. Score-plot visualization of individual exposures displayed a general separation between DP2050 and the remaining sites along the first PLS-R component axis (Figure A9). The PLS-R models generated using the WSOC-normalized component fluorescence ($C1_{WSOC}$–$C3_{WSOC}$) as Y variables were comparatively weak, as a two-component model displayed low cumulative $R_2Y$ (0.48) and $Q_2Y$ (0.18) scores.



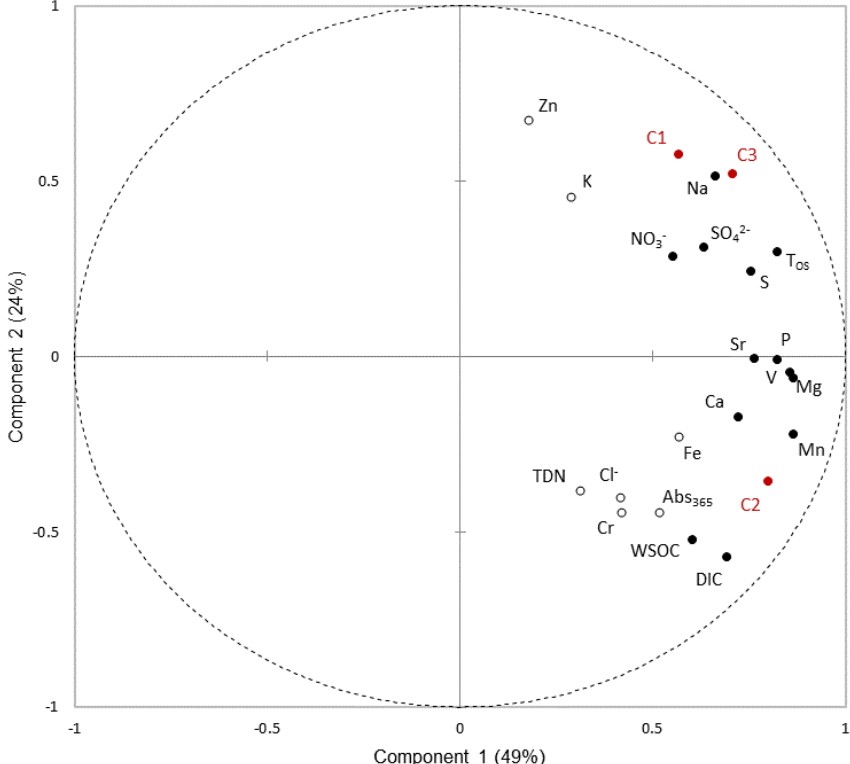

**Figure 6: Correlation biplot displaying both X (black) and Y (red) variables in relation to the first two partial least square regression components. Filled and outlined black points indicate X variables assigned high (≥ 1) and low (< 1) VIP scores, respectively.**

## 4 Discussion

### 4.1 WSOC and WS-BrC absorption

Ambient WSOC in the AOSR (including AMS18) were higher than TSP-based (summertime) measurements (WSOC < 1 µg C m$^{-3}$) from the remote southern Tibetan Plateau (Cong et al., 2015; Li et al., 2021) and Greenland Ice Sheet (Hagler et al., 2007), but similar to the magnitude of values (1–10 µg C m$^{-3}$) reported from Sinhagad, India (Kirillova et al., 2013), Bangkok, Thailand (Tang et al., 2021), Karachi, Pakistan (Chen et al., 2020), and peatland-fire affected samples from west Russia (Popovicheva et al., 2019). Conversely, WSOC in the AOSR was generally lower than the measured concentrations (> 10 µg C m$^{-3}$) in New Delhi, India (Kirillova et al., 2014; Miyazaki et al., 2009).

Compared to previous studies focused on the TSP fraction, Abs$_{365}$ in the AOSR was generally higher than summertime values reported from remote regions including the Tibetan Plateau (Abs$_{365}$ = 0.4 m$^{-1}$, Zhu et al., 2018), Gobi Desert (1.9 m$^{-1}$, Wen et al., 2021), and the Amazon Basin (< 1.4 m$^{-1}$, Saturno et al., 2018), similar to observations from Bangkok (4.5 m$^{-1}$, Tang et al., 2021), and smaller relative to polluted urban areas in Eastern China (8.4 m$^{-1}$, Wen et al., 2021). Observed MAE$_{365}$ in the AOSR was 25 – 400 % higher than the values reported across a diversity of studies (Srinivas and Sarin, 2013; Tang et al., 2021; Wu et al., 2019), demonstrating WS-BrC possessed relatively high light absorption capacity.





The moderate positive correlation between WSOC and $Abs_{365}$ indicated that WS-BrC was an appreciable constituent of soluble organic carbon in the study region, and both TSP fractions likely shared select environmental sources. Higher WSOC and $Abs_{365}$ values at DP2050 relative to the other study sites indicated industrial operations may contribute to regional WS-BrC; the significant positive correlation between $Abs_{365}$ and $T_{OS}$ further supported this assertion. However, the similarity in WSOC and $Abs_{365}$ between the remaining stations (excluding DP2050) implied that the extent of OS influence was spatially limited, and regional sources were

the primary determinant of BrC at further-field locations. Elevated WSOC and absorbance during smoke events (e.g., AMS18, August 5–7) suggested biomass combustion was a source of BrC. Regional wildfire smoke could explain why TSP-phase WSOC and $Abs_{365}$ at the background site (AMS18) was higher than other remote areas such as the Tibetan Plateau and Greenland ice sheet (Hagler et al., 2007; Zhu et al., 2018), but comparable to WSOC concentrations during boreal fire events in west Russia (Popovicheva et al., 2019). Insignificant differences in $MAE_{365}$ between AOSR sites, and inconsistent $MAE_{365}$ values during

extreme fire smoke events demonstrated that aerosol light absorption capacity was not dependant upon OS or wildfire sources.

### 4.2 C1 and C3 fluorescent components

Elevated C1 and C3 fluorescence among near-field (< 5 km) industrial sites (DP2050, AMS01), and significant positive correlations with $T_{OS}$ and continuous pollutant data indicated that OS emissions contributed to measured fluorescence. Depressed $F_{MAX}$ values during exposures where air masses originated from non-OS (AMS01; August 21–23) and/or background regions (i.e.,

AMS18) further indicated that industrial operations were the primary source of C1 and C3 in the AOSR. Moreover, low emission intensity (with exception of C2) during smoke events (DP2050, July 19-20; AMS18, August 5-7; Figure A7) demonstrated that biomass combustion was not the primary source of C1 and C3 fluorescence.

Source-apportionment studies have identified bitumen excavation and hauling as a key source of PM to the AOSR; emissions that are often associated with Na, S, and V (Landis et al., 2012; 2017). Therefore, the importance of Na and $SO_4^{2-}$ as predictors of C1

and C3 variation could further link bitumen emissions to observed fluorescence. The spectral similarity between the raw bitumen EEM and PRLIS C3 confirmed that OS material likely contributed to at least one of the observed fluorescent peaks. Resuspended oil sands materials have been shown to significantly contribute to atmospheric loadings of total PAC species (Landis et al., 2019), where somewhat water-soluble and low molecular weight (LMW) PACs (i.e., naphthalene, fluorene, dibenzothiophene) could fluoresce in the low excitation and emission wavelength range like C1 (Alostaz et al., 2006). Photodegradation of exposed

bituminous material can produce oxygenated PACs (Yang et al., 2016), which due to their enhanced water solubility could additionally contribute to WS-BrC fluorescence.

Fugitive dust emissions from tailings ponds are an additional source of PM in the study region (Landis et al., 2012; 2019; Zhang et al., 2016). Owing to the application of NaOH to accelerate the crude extraction process (Bakhtiari et al., 2015), OS tailings contain high concentrations of Na (MacKinnon and Sethi, 1993; Wang et al., 2015). The strong association of Na with C1 and C3

(via PLS-R) provided evidence to suggest tailings dust contributed to observed fluorescence. Oil sands tailings dust has been identified as a source of 1–3 ring PACs and naphthenic acids (Landis et al., 2012; Yassine and Dabek-Zlotorzynska, 2017; Zhang et al., 2016), which can fluoresce in $\lambda_{Ex}$ and $\lambda_{Em}$ regions like C1 and C3 (Alostaz et al., 2006; Kaur et al., 2014). The spectral similarity between C3 and MFT reference material, and elevated C1 and C3 emission intensity during exposures downwind of tailings-sites further implied fugitive facility emissions contributed to BrC fluorescence.

Emission source profiles previously identified in the AOSR have been shown to contain loadings indicative of both crustal (Mg, Ca, Mn) and petrogenic origin (S, V, Ni) (Landis et al., 2012), largely because of the proximity and co-emission of different dust sources associated with OS activities. The first PLS-R component, which displayed strong correlations with Mg, Ca, Mn, C1, and





C3, may be broadly representative of crustal emissions (e.g., haul road dust, excavated surface-soils and overburden) from OS operations. However, separation of Mg, Ca, and Mn from C1 and C3 along the second PLS-R component (non-crustal) axis plot implied the fluorophores were co-emitted along-side these crustal sources.

Analysis of the current TSP dataset was unsuccessful in distinguishing between C1 and C3, likely indicating the co-emission of these fluorophores from industrial sources. However, unlike C3, the C1 HULIS fluorophore did not match the EEM spectra produced from the raw bitumen or MFT extracts, which may suggest the two fluorophores originated from different emission sources associated with OS operations. Industrial facilities in the AOSR often contain stockpiles and beaches comprised of petroleum coke (petcoke), a carbon-rich derivative of the oil refining process that has been previously identified as a major source of particulate PAC emissions to the surrounding region (Zhang et al., 2016). Fugitive petcoke dust co-emitted from OS facilities could contribute to observed C1 fluorescence.

### 4.3 C2 fluorescent component

Significant positive correlations between C2, WSOC and Abs$_{365}$ suggested that the fluorophore was a key constituent of bulk WS-BrC in the AOSR, consistent with the observed dominance of C2 over total sample fluorescence. Significantly elevated C2 at the near-field (< 1 km) Mildred Lake site, yet comparable fluorescence at AMS18 relative to the remaining stations suggested both industrial and broad regional sources contributed to BrC across the study area, consistent with what was observed for WSOC and Abs$_{365}$.

Occasionally elevated fluorescence during regional smoke events (e.g., July 19-20, DP2050) could indicate wildfire smoke contributed to C2 fluorescence; this is unsurprising given that biomass combustion is one of the leading sources of BrC globally (Chen and Bond, 2010). Particulate matter originating from biomass combustion can contain substantial quantities of water-soluble organic carbon (Park et al., 2013; Saarnio et al., 2010), which could explain the observed association between WSOC and C2.

The similarity in fluorescent spectra when comparing C2 and road dust EEMs indicated the potential influence of crustal fugitive dust emissions over fluorescent PM. Haul roads throughout the AOSR are generally constructed using locally excavated limestone and bitumen, and regional road dust emission profiles often contain high loadings of DIC, Mg, Ca, and Mn (Wang et al., 2015). The observed associations (PLS-R) between these variables and C2 among the TSP samples further supported the potential influence of road dust emissions over measured fluorescence. Road dust materials can contain a diversity of chromophoric organic matter, including amino acids (Chalbot and Kavouras, 2019) and LMW PACs (in the case of hauling and industrial roads in the OS) (Landis et al., 2019) that can fluoresce in the low $\lambda_{Ex}$ and $\lambda_{Em}$ wavelength range like C2 (Christensen et al., 2005; Kim and Koh, 2020). Following initial suspension, larger diameter road dust particles are rapidly removed from the atmosphere through physical interactions with the surface environment (vegetation, man-made structures, etc.). For example, Veranth et al. (2003) found that irregular surface conditions adjacent to an unpaved road in Utah (United States) contributed to an 85 % reduction of coarse particulate matter within 100 m of the emission source. In the case of the AOSR, the proximity (< 1 km) of site DP2050 to OS haul roads could contribute to elevated C2 fluorescence, while rapid removal further downwind may explain why C2 was relatively similar between the remaining downfield sites. In the absence of appreciable OS road dust influence, regional wildfire smoke likely becomes the dominant source of C2 fluorescence.



## 5 Conclusions

Optical analysis of atmospheric TSP samples collected throughout the AOSR found that OS activities were a measurable source of WS-BrC to the surrounding airshed during the summer season. Combined EEM and PARAFAC evaluation of the TSP identified three distinct fluorophores (C1–C3), all of which exhibited a potential link to industrial activity. However, uniformly elevated C2 emission intensity during wildfire smoke episodes indicated biomass combustion was also a considerable source of PRLIS aerosols to the region, while C1 and C3 were linked predominantly to anthropogenic OS sources. The added emissions from local OS industry could compound the environmental and human health impacts (e.g., planetary albedo, respiratory distress) of wildfire smoke WS-BrC, which are expected to worsen into the future partially as a result of climate change.

As far as the authors are aware, this is the first study to evaluate WS-BrC in the AOSR airshed. Considering the low operational costs compared to conventional methods of mass spectrometry, UV-Vis and EEM-PARAFAC analysis offer accessible techniques to monitor WS-BrC and initially screen for OS-sourced aerosols throughout the AOSR. Moreover, the capacity of EEM-PARAFAC to identify fluorescent species strongly linked to industry suggested that similar spectroscopic techniques could be employed to evaluate WS-BrC near other oil and gas facilities worldwide.

## Appendix A

Datasets were acquired using an Agilent 8800 triple quadrupole inductively coupled mass spectrometer, in the Water Quality Centre at Trent University. A MicroMist nebulizer (nominal uptake rate 400 $\mu$L min$^{-1}$) and Scott double pass spray chamber was used for sample introduction.

**Table: Instrument operating conditions and measurement parameters.**

| | |
|---|---|
| RF power | 1550 W |
| Carrier gas flow rate[a] | 1.05 L min$^{-1}$ |
| Sampling depth[a] | 10 mm |
| Extraction lens | X-type |
| Sampler and skimmer cone | Standard nickel |

[a] General Purpose Plasma Preset

Calibration standards were prepared by serial dilution of a 100 ng mL$^{-1}$ multi-element solution (HPS QCS 27) and 1000 ng mL$^{-1}$ single element standards in 2% HNO$_3$. High purity water (18.2 M$\Omega$) and nitric acid was used for the preparation of all solutions. The NIST SRM 1640a (Trace Elements in Natural Water), NRC SLRS-6 (River Water Certified Reference Material for Trace Metals) and CALA PT standards were used for QA/QC. The measured concentrations were within 5% of the certified values.

**Table: Instrument tune mode and scan type used to measure each element.**

| Element | Tune Mode | Scan Type |
|---|---|---|
| Na | He | Single Quad |
| Mg | He | Single Quad |
| P | O$_2$ | MS/MS |
| S | O$_2$ | MS/MS |
| K | H$_2$ | MS/MS |
| Ca | H$_2$ | MS/MS |
| V | O$_2$ | MS/MS |
| Cr | He | Single Quad |
| Mn | He | Single Quad |
| Fe | H$_2$ | MS/MS |
| Zn | He | Single Quad |
| Sr | No gas | Single Quad |




**Table A1. Spearman correlation analysis comparing absorbance indices (Abs$_{365}$, MAE$_{365}$), modelled fluorescent components (C1-C3), trajectory frequencies (T$_{OS}$), and continuous pollutant species measured at the active sampling locations (excluding 2050). The correlation coefficient (R$_S$) for each significant correlation (α < 0.05) is shown.**

| | T$_{OS}$ | THC | TRS | SO$_2$ | NO$_2$ | NO | CO | PM$_{2.5}$ |
|---|---|---|---|---|---|---|---|---|
| THC | 0.87 | - | - | - | - | - | - | - |
| TRS | 0.92 | 0.85 | - | - | - | - | - | - |
| SO$_2$ | 0.84 | 0.73 | 0.89 | - | - | - | - | - |
| NO$_2$ | 0.9 | 0.91 | 0.88 | 0.82 | - | - | - | - |
| NO | 0.82 | 0.83 | 0.89 | 0.82 | 0.89 | - | - | - |
| CO | 0.5 | 0.47 | 0.63 | 0.65 | 0.66 | 0.54 | - | - |
| PM$_{2.5}$ | 0.48 | 0.57 | 0.5 | 0.42 | 0.6 | 0.4 | 0.67 | - |
| C1 | 0.77 [a] | 0.7 | 0.67 | 0.54 | 0.66 | 0.57 | 0.48 | 0.72 |
| C2 | 0.62 [a] | - | - | - | 0.42 | 0.42 | - | - |
| C3 | 0.88 [a] | 0.73 | 0.73 | 0.6 | 0.71 | 0.65 | 0.4 | 0.58 |
| MAE$_{365}$ | - | - | - | - | - | - | - | - |
| Abs$_{365}$ | 0.48 [a] | - | - | - | - | 0.36 | - | - |


[a] Correlation calculated using data from all stations, including DP2050.

**Table A2: Spearman correlation analysis of modelled fluorescent components (C1-C3) against corresponding chemical and optical variables measured through the active sampler network. The coefficient (R$_S$) of each significant correlation (α < 0.05) is shown.**


| Analyte | PARAFAC | | | Component / WSOC | | | Absorbance | |
|---|---|---|---|---|---|---|---|---|
| | C1 | C2 | C3 | C1$_{wsoc}$ | C2$_{wsoc}$ | C3$_{wsoc}$ | Abs$_{365}$ | MAE$_{365}$ |
| WSOC | - | 0.70 | - | - | - | - | 0.55 | - |
| DIC | - | 0.80 | - | - | - | - | 0.57 | - |
| TDN | - | 0.44 | - | - | - | - | 0.39 | - |
| NO$_3^-$ | - | 0.49 | 0.42 | - | - | - | - | - |
| SO$_4^{2-}$ | 0.54 | 0.48 | 0.58 | 0.41 | - | 0.40 | - | - |
| Cl$^-$ | - | - | - | - | - | - | - | - |
| Na$^+$ | 0.62 | 0.61 | 0.75 | 0.57 | - | 0.60 | 0.36 | - |
| Mg$^{2+}$ | 0.49 | 0.62 | 0.66 | 0.41 | - | 0.46 | - | - |
| K$^+$ | - | - | - | - | - | - | - | - |
| Ca$^{2+}$ | 0.35 | 0.74 | 0.57 | - | - | - | 0.54 | - |
| P | - | 0.71 | 0.55 | - | - | - | 0.55 | - |
| S | 0.62 | 0.56 | 0.71 | 0.65 | 0.36 | 0.68 | - | - |
| V | 0.52 | 0.65 | 0.69 | 0.39 | - | 0.45 | 0.52 | - |
| Cr | - | 0.72 | - | - | - | - | 0.57 | - |
| Mn | 0.37 | 0.63 | 0.57 | - | - | - | - | - |
| Fe | 0.36 | 0.40 | 0.33 | - | - | - | 0.42 | - |
| Zn | - | - | - | - | - | - | - | - |
| Sr | 0.62 | 0.63 | 0.79 | 0.56 | 0.47 | 0.60 | 0.42 | - |
| Abs$_{365}$ [a] | - | 0.74 | 0.37 | - | - | - | - | 0.63 |
| MAE$_{365}$ [b] | - | - | - | - | 0.57 | 0.36 | 0.63 | - |

[a] Reported as m$^{-1}$
[b] Reported as m$^{-2}$gC$^{-1}$



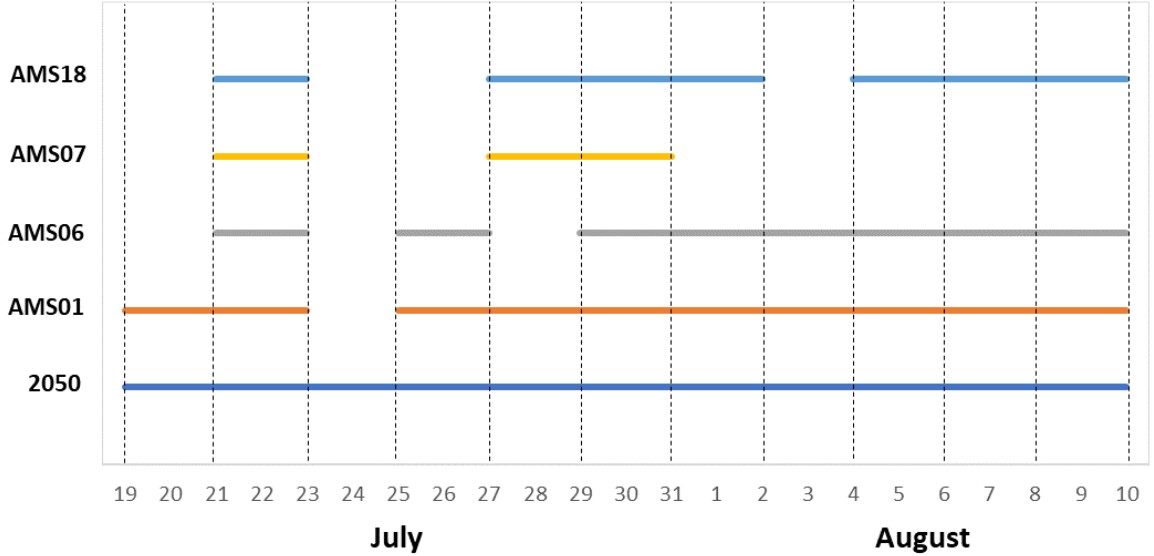


**Figure A1: Temporal distribution of active sampler exposures (demarked by the vertical dotted lines) at each of the WBEA monitoring sites during the field study (July 19–August 10, 2023).**






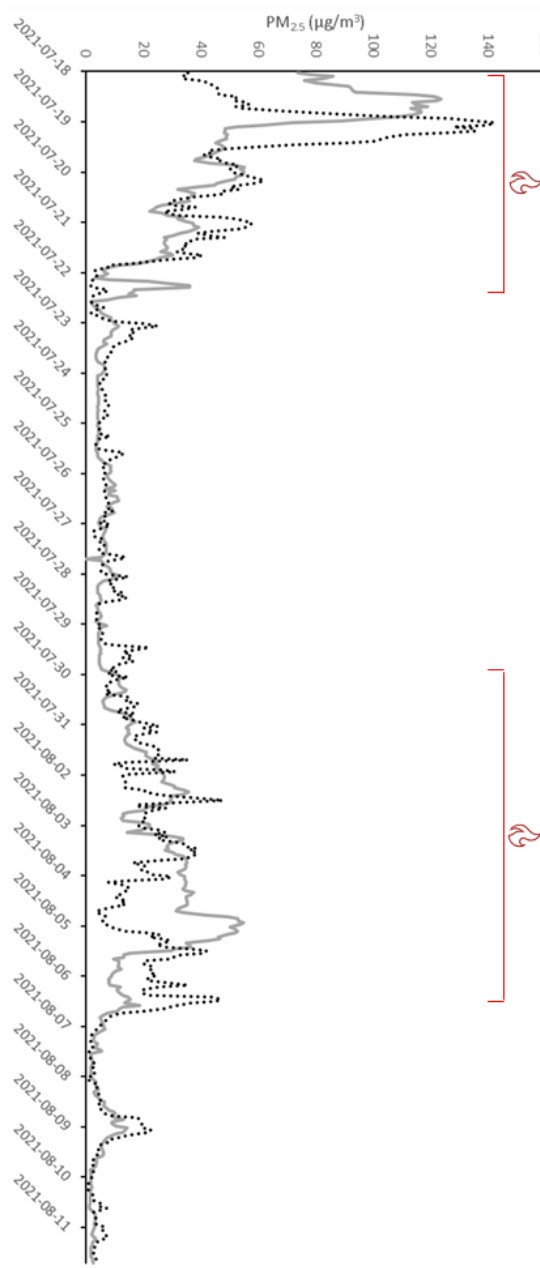

**Figure A2: Temporal variation of average hourly PM$_{2.5}$ (µg m$^{-3}$) measured at a) AMS01 (black dotted line) and b) AMS18 (grey line) continuous monitoring stations during the study period (July 19–August 10, 2021). Approximated wildfire smoke events have been**
**outlined in red.**



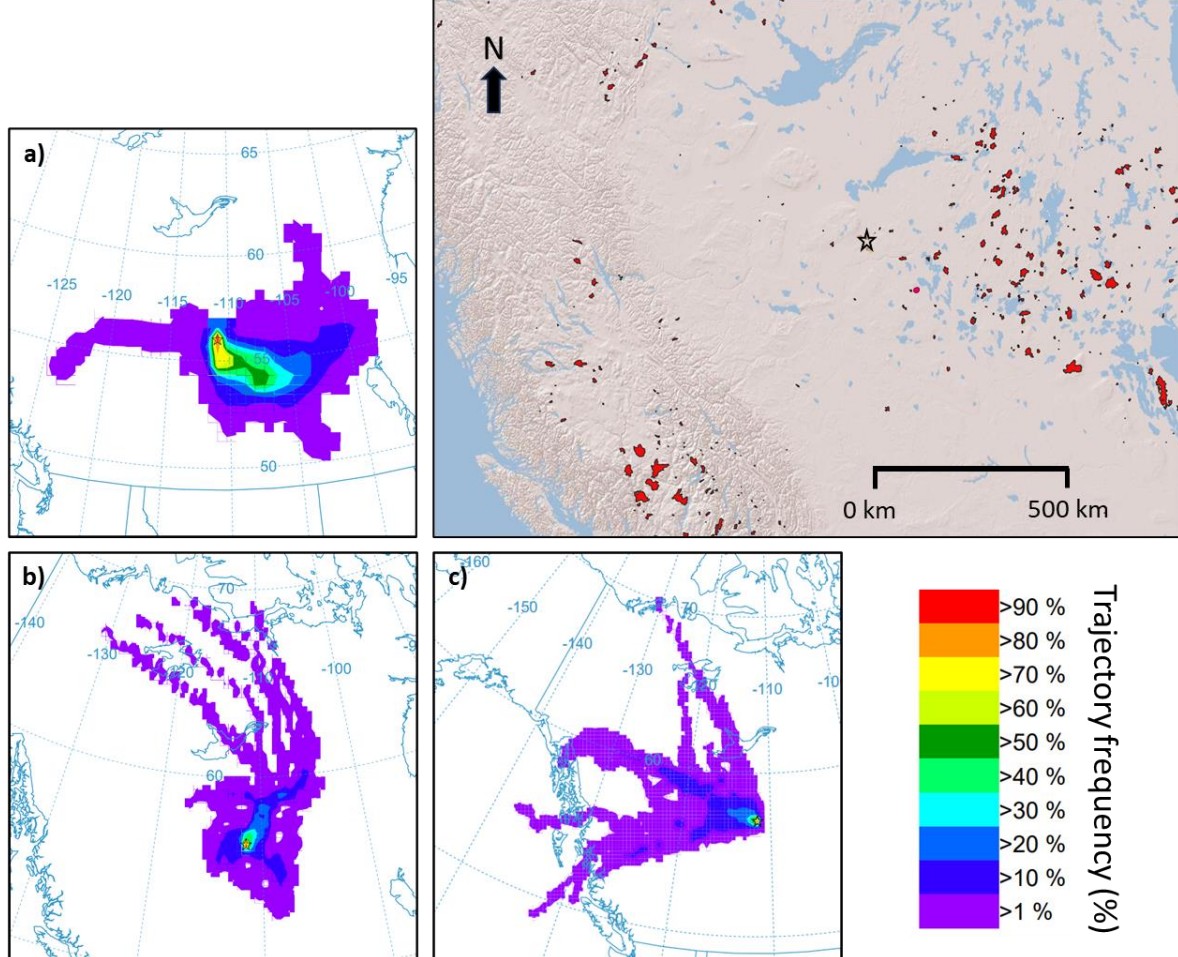

**Figure A3: Back-trajectory (48 h) frequency (%) plots displaying atmospheric transport to AMS01 (black outlined star) during high PM$_{2.5}$ periods a) July 19 - 22, b) July 31 – August 7), and the low PM$_{2.5}$ period, c) July 23 – 30. The top-right map displays the spatial distribution of wildfire-affected areas (red polygons) throughout western Canada during the study period (July 19 – August 10) (© ESRI 1995-2024). Note that the spatial scale of the trajectory plots and wildfire map are not equal.**





**Figure A4: Comparison of HYSPLIT 48 h back-trajectories and continuous wind direction measurements corresponding to the July 25-27 and August 6-8 exposures at AMS01 (a & b) and AMS06 (c & d) continuous monitoring stations, respectively (© ESRI 1995-2024).**








**Figure A5: Spatial distributions of 48 h atmospheric back-trajectories (black lines) converging at sampling locations during selected exposure periods, including: a) AMS18; August 7-9, b) AMS06; July 21-23, c) DP2050; July 19-20, and d) AMS18; August 5-7. Normalized $F_{MAX}$ values of C1 – C3 (QSU m$^{-3}$) measured during each exposure are shown beneath**

**the corresponding trajectory plots (© ESRI 1995-2024).**



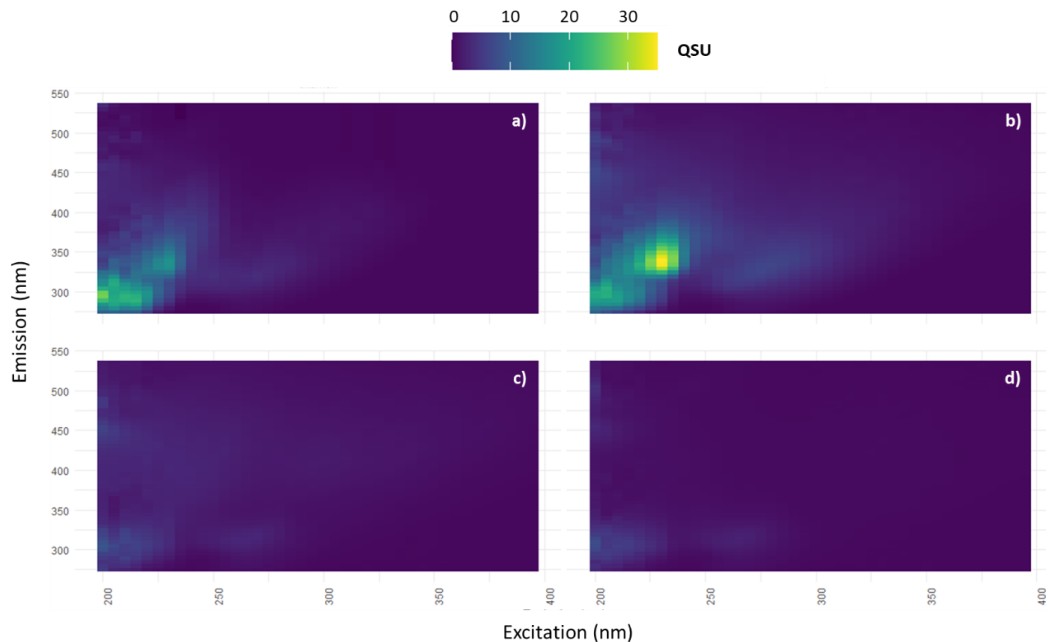

**Figure A6: Visualization of the EEM scans produced from a) raw bitumen, b) dry MFT, and unpaved road dust from c) DP2050 and d) AMS01. Fluorescence intensity is displayed in QSU.**

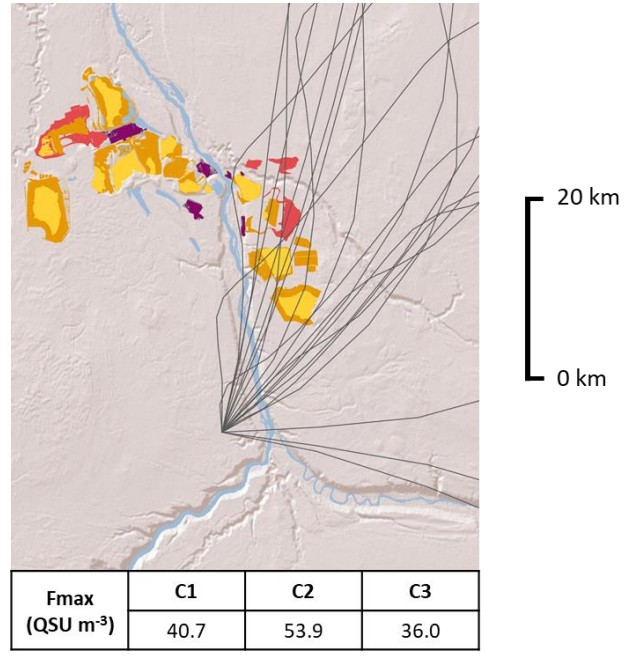

| Fmax (QSU m$^{-3}$) | C1 | C2 | C3 |
|---|---|---|---|
| | 40.7 | 53.9 | 36.0 |


**Figure A7: Spatial distributions of 48 h atmospheric back-trajectories (black lines) converging at AMS06 during the August 4–6 exposure. $F_{MAX}$ values of C1–C3 (QSU m$^{-3}$) measured during the corresponding exposure are shown. The spatial boundaries of various OS facilities, including wet tailings (yellow), dry tailings (orange), open mine pits (red), and plant sites (purple) are shown (© ESRI 1995-2024).**






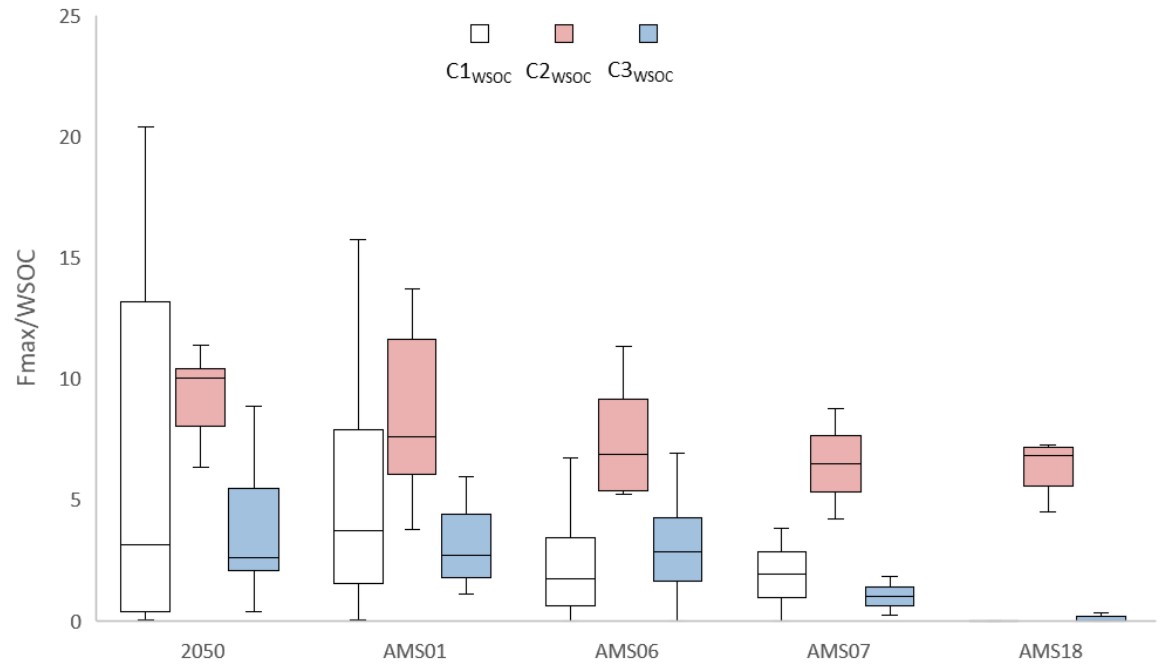

**Figure A8: Distributions of WSOC - normalized component fluorescence (C1$_{WSOC}$ – C3$_{WSOC}$) measured at each sampling location during the study period (July 19 – August 10, 2021).**




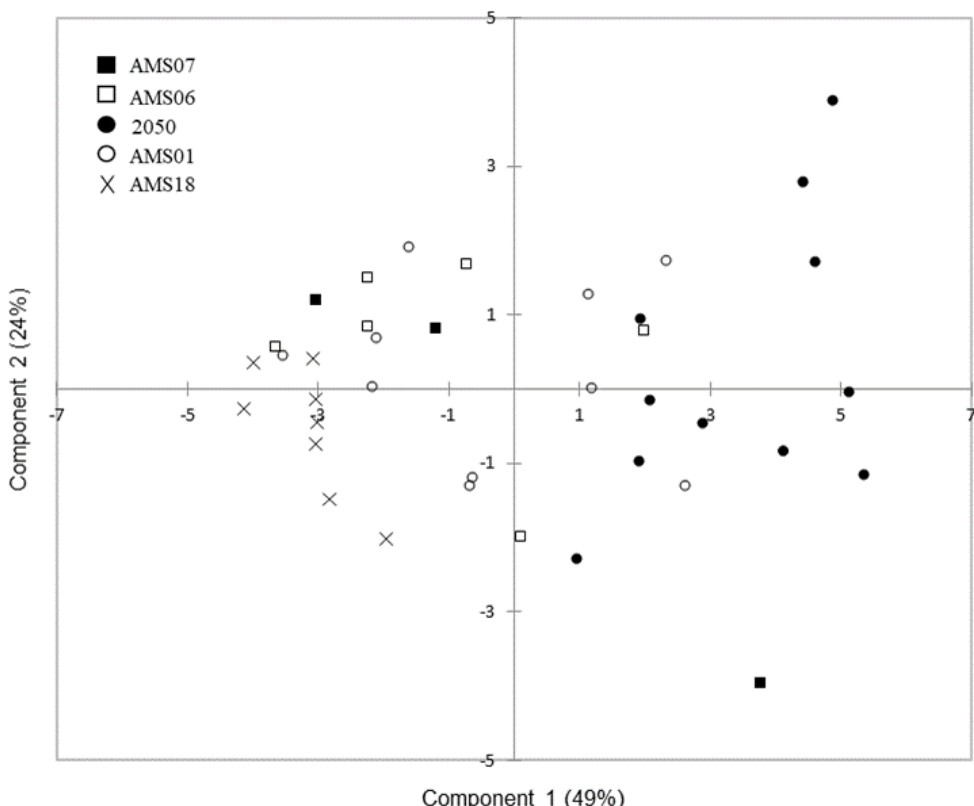


**Figure A9: PLS-R Biplot displaying the distribution of observations relative to the first two modelled components. Observations are categorized according to site ID, including: AMS01 (open circle), DP2050 (full circle), AMS06 (open square), AMS07 (full square), AMS18 (x-mark).**

**Author contributions**

Conceptualization: D.B., J.A.; Field sampling and laboratory analysis: D.B., M.G., D.H.D., J. A.; Writing original draft: D.B.; Writing review and editing: D.B., M.G., D.H.D., P.A.M., J. A

**Competing interests**

The authors declare that they have no conflict of interest.


**Acknowledgements**

This study was made possible through collaboration with the Wood Buffalo Environmental Association (Alberta, Canada), laboratory analysis by the Trent University Water Quality Center (Ontario, Canada), and research funding provided by Environment and Climate Change Canada (GCXE20S046). The primary author (D.B.) would also like to thank Mark Gordon
and Timothy Jiang for their assistance in the field.

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
