# Peer review of "Characterization of atmospheric water-soluble brown carbon in the Athabasca Oil Sands Region, Canada."

_EGUsphere, 2024_

## Referee Comment (RC2)

Review of 'Characterization of atmospheric water-soluble brown carbon in the Athabasca Oil Sands Region, Canada' by Blanchard et al.

General comments:

The study investigated water-soluble brown carbon (WS-BrC) in the Athabasca Oil Sands Region, Alberta, Canada, using filter-pack sampling and spectroscopic techniques. EEM-PARAFAC analysis identified three fluorescent components, with humic-like (C1) and protein-like (C2, C3) fluorophores linked to oil sands emissions and wildfires. Increased C1 and C3 fluorescence near industrial sites indicated their origin from oil sands activities. C2's high intensity near industrial sites and during wildfires suggested both wildfire-generated and anthropogenic sources. The study demonstrated EEM-PARAFAC's effectiveness in monitoring industrial WS-BrC. However, some comments should be addressed to improve the study.

Specific comments:

1) My main concern is the innovative points of this article. If it emphasizes the impact of industrial operation on this region, the data support of the paper is insufficient. If it is the impact of wildfires, the selection of background sites seems to be inappropriate. If it is the application of monitoring technology, it may also lack validation of other methods. So, you make a revision to clarify the innovative of the paper.

2) Line 108-111: The number of total samples are different in this two sentences. Please check it carefully.

3) Line 195: Misspelling of "Reference".

4) Line 227-230: Can relative frequency data accurately reflect the impact of industrial facilities on fluorescence, as intersection cannot indicate the emissions and substances emitted by industrial facilities, but only indicates spatial overlap.

5) Table 2: From the MAE365 data, there is no significant difference between the AMS18 site and other sites affected by the oil sands industry. Therefore, it is necessary to consider 1) whether the selection of background sites is reasonable,

or 2) the impact of the oil sands industry on water-soluble BrC from other perspectives, because we know that an important aspect of discussing brown carbon is its light absorption capacity.

6) Line 295-296: "Similar scans of unpaved road dust and sand pit materials displayed comparatively weak fluorescence ($\leqslant$ 8 QSU) within these EEM regions". However, it can be obviously observed in Fig A6. Maybe, you can revise the fig A6c,d to make it more obvious.

7) Line 302-310: I also have concerns on the data processing of EEM: In fact, the three fluorescent components are not relevant with humic-like and protein-like substances that derived from water (phytoplankton)/terrestrail soil, but they should are resided on the same peak positions. Author should give some explanation like this way using references, otherwise, some readers can mistake to have similarly or dissimilarly.

8) Line 416-418: Can certain biomarkers be used to establish this intermediate association? Just like using levoglucosan to represent the contribution of combustion sources, as introduced in the introduction, some chemicals such as polycyclic aromatic compounds, oxygenated polycyclic aromatic hydrocarbons, and naphthonic acids have been found in organic particles produced by oil sands (Introduction, Line 42).

---

## Author Comment (AC2)

**Reviewer #1**

**General comments**

This study investigated the WS-BrC from AOSR industrial operations by using the ultraviolet-visible spectroscopy, fluorescence excitation-emission matrix (EEM) spectroscopy, and the Parallel Factor Analysis (PARAFAC). The analysis identified three fluorescent components: one humic-like (C1) and two protein-like (C2, C3). This study indicated that the Increased fluorescence of C1 and C3 was linked to proximity to oil sands facilities and correlated with pollutants indicative of industry emissions. Component C2 was prominent during wildfire smoke events and associated with anthropogenic sources as well. This study proposed that combined EEM-PARAFAC analysis would be an effective and accessible method for monitoring WS-BrC in the AOSR. Overall, this paper is within the scope of ACP and would be of great interest to the broad atmospheric science community. I recommend that this paper can be accepted after minor revision.

Response: Thank you for the recommendation.

My main comment is to explicitly highlight the significance and novelty of this study in the context of climate change. Section 1, 2.1, and 4.1 collectively imply that, given the large area of AOSR and the high BrC light absorption capacity, WS-BrC in this area may exhibit indispensable influence on the global climate. Can the authors summarize the information and explicitly discuss the potential global impact in the Conclusion section?

Response: As suggested, we have added additional context in section 1 to highlight the significance and novelty of our study in the context of climate change; further, we have discussed the potential impact of OS emitted WS-BrC over the climate in the conclusion (section 5). Although industrial emissions appeared to impact aerosol absorbance within the immediate environment (as demonstrated by $Abs_{365}$ at DP2050), statistically similar $Abs_{365}$ among the remaining sites suggested OS sourced WS-BrC exerted limited influence over the regional climate during the summer.

Lines 471-475: "Significantly higher $Abs_{365}$ measured at the industry adjacent site (DP2050) demonstrated that OS operations likely enhanced aerosol light absorbance, which in turn could impact local solar radiation budgets and atmospheric photochemistry. However, statistically similar $Abs_{365}$ and $MAE_{365}$ values across the remaining stations indicated that the influence of OS emissions over WS-BrC absorbance was spatially limited and unlikely to substantially influence the regional climate during the summer."

**Specific comments:**

1) Please clarify why the pH of sample extracts was adjusted to ~6.5 prior to the optical analysis.

Response: The optical properties of BrC (and CDOM, broadly) are sensitive to environmental pH (for instance, shifting pH can induce a fluorescent quenching effect). In our study, the sample extracts were adjusted to a constant pH to limit the influence of variable pH over absorbance and fluorescence. We have added a brief rational for this adjustment in the MS.

2) Please briefly explain the principle of how the fluorescence analyses combined with the PARAFAC can identify WS-BrC in the first paragraph of Section 2.4.

Response: We have included a brief explanation in section 2.4.

Lines 156-158: "Combined EEM and PARAFAC analysis is commonly used to evaluate the optical and structural characteristics of environmental CDOM since the modelling technique can identify the excitation and emission peaks of multiple distinct fluorophores (i.e., HULIS, PRLIS) within EEM scans."

3) Line 195: "refer1ence" should be "reference".

Response: Thank you. Error corrected.

4) Lines 308-310: Does this information suggest the uniqueness of organic aerosol composition within the oil sands region?

Response: Good question — in brief, no. The absence of similar models is most likely because the OpenFluor database contain very little (if any) BrC data. Alternatively, the AOSR fluorophores are spectrally similar to other BrC fluorophores reported in the literature (outside of OpenFluor).

We originally decided to compare the WS-BrC spectra C1-C3 against OpenFluor because the database contained PARAFAC models generated from surface-water samples contaminated with crude oil material. We have decided to remove the OpenFluor portion from the MS and in its place have included comparison to additional BrC studies.

5) Lines 456-463: According to the information in this paragraph, can the authors discuss what kind of further research is required to understand the more detailed emissions profiles of C1 and C3 in relation to regional industrial activities?

Response: We have included a sentence expanding on this matter.

Lines 440-442: "Future BrC field studies should consider increased sampling locations near a diverse range of OS facilities, shortened exposure periods (to limit source mixing), and expanded analysis of complimentary OS indicator variables (e.g., molybdenum, nickel, PACs, [Landis et al., 2019]) to better characterize C1 and C3 source profiles in the AOSR."

6) Lines 485-486: Is there any approach to quantify the contribution of wildfires to the C2 fluorescence?

Response: Good question. Certain organic compounds like levoglucosan and retene are frequently used in the literature as pyrogenic tracers that can assist with wildfire source apportionment. Unfortunately, we did not measure these indicator species from the TSP filter extracts. We have added discussion in section 4.3 recommending the inclusion of these tracer species in future BrC studies in the AOSR.

Lines 453-454: "Future evaluations of BrC in the AOSR should additionally measure pyrogenic tracer species such as retene and levoglucosan, as these organic compounds can be used for wildfire source apportionment (Wentworth et al., 2018)."
* * *
**Reviewer #2**

**Review of 'Characterization of atmospheric water-soluble brown carbon in the Athabasca Oil Sands Region, Canada' by Blanchard et al.**

**General comments:**

The study investigated water-soluble brown carbon (WS-BrC) in the Athabasca Oil Sands Region, Alberta, Canada, using filter-pack sampling and spectroscopic techniques. EEM-PARAFAC analysis identified three fluorescent components, with humic-like (C1) and protein-like (C2, C3) fluorophores

linked to oil sands emissions and wildfires. Increased C1 and C3 fluorescence near industrial sites indicated their origin from oil sands activities. C2's high intensity near industrial sites and during wildfires suggested both wildfire-generated and anthropogenic sources. The study demonstrated EEM-PARAFAC's effectiveness in monitoring industrial WS-BrC. However, some comments should be addressed to improve the study.

Response: Thank you. We appreciate your interest in the study and constructive feedback.

**Specific comments:**

1) My main concern is the innovative points of this article. If it emphasizes the impact of industrial operation on this region, the data support of the paper is insufficient. If it is the impact of wildfires, the selection of background sites seems to be inappropriate. If it is the application of monitoring technology, it may also lack validation of other methods. So, you make a revision to clarify the innovative of the paper.

Response: As suggested, we have highlighted the significance and novelty of our study (also suggested by Reviewer 1). The primary objective of this work was to determine whether OS industry was a source of WS-BrC in the surrounding airshed. We found that local industry measurably contributed to BrC, i.e., there was: a) significantly elevated $abs_{365}$ and fluorophore intensity among industry adjacent sites; b) positive agreement between fluorophores (C1, C3) and industrial indicator variables (dissolved metals, $SO_4^{2-}$, TRS, $NO_2$, $SO_2$); c) elevated $abs_{365}$ and fluorescence when samplers received atmospheric transport from the direction of OS facilities, and; d) the spectral consistency between C1-C3 and OS material extracts (raw bitumen and mature fine tailings). We believe that our approach is novel to the AOSR and indicates the impact of industrial operations on the region (in the context of air quality and climate change).

2) Line 108-111: The number of total samples are different in this two sentences. Please check it carefully.

Response: Error corrected. Thank you for highlighting this mistake.

3) Line 195: Misspelling of "Reference".

Response: Error corrected. Thank you.

4) Line 227-230: Can relative frequency data accurately reflect the impact of industrial facilities on fluorescence, as intersection cannot indicate the emissions and substances emitted by industrial facilities, but only indicates spatial overlap.

Response: Good point. Yes, the frequency data ($T_{OS}$) is not representative of facility emissions; however, the trajectory analysis served as a representation of atmospheric transport and potential source regions during each exposure, and roughly indicated the likelihood that air sampled during an exposure was influenced by industry. The strong positive correlations between $T_{OS}$ and co-located measurements (via continuous sampler) of industrial pollutants ($SO_2$, TRS, $NO_2$, …) supported this interpretation of the frequency data.

5) Table 2: From the $MAE_{365}$ data, there is no significant difference between the AMS18 site and other sites affected by the oil sands industry. Therefore, it is necessary to consider 1) whether the selection of background sites is reasonable, or 2) the impact of the oil sands industry on water-soluble BrC from other perspectives, because we know that an important aspect of discussing brown carbon is its light absorption capacity.

Response: Yes, the insignificant difference in MAE$_{365}$ between stations suggested that OS emissions did not influence the light absorption efficiency per unit WSOC. This limited influence is consistent with the observed abs$_{365}$ values between stations, where OS emissions only appeared to impact to WS-BrC absorbance at fence-line locations. We have made note of this in the MS (sections 4.1 and 5).

Lines 471-475: "Significantly higher Abs$_{365}$ measured at the industry adjacent site (DP2050) demonstrated that OS operations likely enhanced aerosol light absorbance, which in turn could impact local solar radiation budgets and atmospheric photochemistry. However, statistically similar Abs$_{365}$ and MAE$_{365}$ values across the remaining stations indicated that the influence of OS emissions over WS-BrC absorbance was spatially limited and unlikely to substantially influence the regional climate during the summer."

6) Line 295-296: "Similar scans of unpaved road dust and sand pit materials displayed comparatively weak fluorescence ($\leq$ 8 QSU) within these EEM regions". However, it can be obviously observed in Fig A6. Maybe, you can revise the fig A6c,d to make it more obvious.

Response: Apologies, we are not certain what is meant by this remark. The emission intensity values displayed in the road dust and sand pit EEMs (Fig. 6c & 6d) fall bellow 8 QSU, as illustrated by the colour scale. However, we have now stated in the MS that the dust extract EEMs displayed weak fluorescence in Ex-Em peak regions observed from the TSP and reference material samples.

7) Line 302-310: I also have concerns on the data processing of EEM: In fact, the three fluorescent components are not relevant with humic-like and protein-like substances that derived from water (phytoplankton)/terrestrail soil, but they should are resided on the same peak positions. Author should give some explanation like this way using references, otherwise, some readers can mistake to have similarly or dissimilarly.

Response: We included the Openfluor analysis largely to compare the WS-BrC spectra (from the AOSR) against PARAFAC models generated from surface-water samples containing crude oil. However, you raise a good point – despite the spectral overlap, these fluorescent compounds have been measured in distinct environmental media. To avoid confusion and misinterpretation of the results, we have removed comparisons to surface-water humic and protein-like substances.

8) Line 416-418: Can certain biomarkers be used to establish this intermediate association? Just like using levoglucosan to represent the contribution of combustion sources, as introduced in the introduction, some chemicals such as polycyclic aromatic compounds, oxygenated polycyclic aromatic hydrocarbons, and naphthonic acids have been found in organic particles produced by oil sands (Introduction, Line 42).

Response: Good question. Yes, there are select species (levoglucosan, retene) that can indicate pyrogenic sources but unfortunately, we do not have access to such data. In our opinion, the available data still offer sufficient evidence that WSOC and Abs$_{365}$ are partially influenced by biomass combustion and industrial emissions, given that: a) WSOC and Abs$_{365}$ were significantly elevated at DP2050 but similar between remaining stations (including the remote site, AMS18), and b) WSOC and Abs$_{365}$ were high during periods of `wildfire smoke transport. We have added discussion in the MS suggesting how future field studies could better distinguish between emission sources.

Lines 440-442: "Future BrC field studies should consider increased sampling locations near a diverse range of OS facilities, shortened exposure periods (to limit source mixing), and expanded analysis of complimentary OS indicator variables (e.g., molybdenum, nickel, PACs, [Landis et al., 2019]) to better characterize C1 and C3 source profiles in the AOSR."

Lines 453-454: "Future evaluations of BrC in the AOSR should additionally measure pyrogenic tracer species such as retene and levoglucosan, as these organic compounds can be used for wildfire source apportionment (Wentworth et al., 2018)."

---

## Author Response (AR2)

**Reviewer #1**

**Specific Comments** :

Some of the references in this manuscript can be updated, Especially, the references early than 2014. It is important to explain the existing problems.

Good point – we have replaced several references with more recent studies. Note that some older references remain in the MS as these sources are foundational to specific methods used in the current study.

Line 456-470: For C2 fluorophore, wildfire event and road dust both contributed to its emission, as well as industry emission. How to quantify the contributions of different sources? That is to say, all three types of emission sources have an impact on C2. So how should their impact be quantified?

Good question – Certain organic compounds like levoglucosan and retene are frequently used as pyrogenic tracers, while specific metals and PACs are used as oil sands indicator species. These tracers could be used to assess the relative contribution of biomass combustion and industrial emissions to C2 fluorescence; however, we did not measure these indicator species from the TSP filter extracts. We have included a statement recommending the use of these indicator variables to determine source contributions in future studies.

Lines 467-469: "Future studies should consider using pyrogenic (retene, levoglucosan) and petrogenic (molybdenum, nickel, PACs) indicator variables to determine the relative contribution of wildfire and OS emission sources to C2 fluorescence in the AOSR."

Line 295-296: "Similar scans of unpaved road dust and sand pit materials displayed comparatively weak fluorescence (≤ 8 QSU) within these EEM regions". However, it can be obviously observed in Fig A6. Maybe, you can revise the fig A6c,d to make it more obvious. The meaning is that the scale of the color scale can be adjusted to make it more prominent.

Thank you for the clarification. We have adjusted the colour scale of the figure in question (now Figure A1c and A1d).